# Genomic and Cytotoxic Damage in Wistar Rats and Their Newborns After Transplacental Exposure to *Hibiscus sabdariffa* Hydroalcoholic Extract

**DOI:** 10.3390/ijms26157448

**Published:** 2025-08-01

**Authors:** Yelin Tobanche Mireles, Ana Lourdes Zamora-Pérez, Marisol Galván Valencia, Susana Vanessa Sánchez de la Rosa, Fuensanta del Rocío Reyes Escobedo, Blanca Patricia Lazalde-Ramos

**Affiliations:** 1Maestría en Ciencias Biomédicas, Área de Ciencias de la Salud, Universidad Autónoma de Zacatecas, Zacatecas 98000, Mexico; yelintomi@gmail.com (Y.T.M.); gavm001144@uaz.edu.mx (M.G.V.); 2Instituto de Investigación en Odontología, Centro Universitario de Ciencias de la Salud, Universidad de Guadalajara, Guadalajara 44340, Mexico; anazamora@gmail.com (A.L.Z.-P.); sanvan0937@gmail.com (S.V.S.d.l.R.); 3Unidad Académica de Ciencias Químicas, Universidad Autónoma de Zacatecas, Zacatecas 98000, Mexico; fuenreyes@uaz.edu.mx

**Keywords:** *Hibiscus sabdariffa*, micronucleus assay, lipid peroxidation, malondialdehyde

## Abstract

*Hibiscus sabdariffa (Hs)* is a tropical plant with a wide range of therapeutic properties; however, few studies have evaluated its potential adverse effects. In the present study, the cytotoxic and genotoxic effects of the hydroalcoholic extract of *Hs* (EH*Hs*) dried calyces administered during gestation were assessed in Wistar rats and their newborns using the micronucleus assay in peripheral blood and the quantification of malondialdehyde (MDA) in various tissues. Three different doses of EH*Hs* (500, 1000, and 2000 mg/Kg) were administered orally to five pregnant Wistar rats per group during the final days of gestation (days 16–20). Blood samples were collected every 24 h during the last six days of gestation and from the neonates at birth, along with tissue samples for MDA quantification. EH*Hs* induced myelosuppression in the mothers and genotoxicity in their newborns, as well as cytotoxicity, evidenced by increased MDA levels in serum, liver, and kidneys of the mothers, and in the liver, kidneys, brain, and muscle tissues of the neonates. These findings provide important insights into the safety profile of *Hs*, and its use is therefore recommended only under the supervision of a qualified healthcare professional.

## 1. Introduction

Although modern medicine is well developed in most parts of the world, large segments of the population in developing countries still rely on traditional medicine for primary healthcare [1,2,3]. Traditional medicine includes the use of herbal extracts to treat or prevent health conditions. Among the plants used for the preparation of herbal extracts is *Hibiscus sabdariffa* (*Hs*), commonly known as “Roselle” or “Jamaica”. This shrub, which belongs to the Malvaceae family, has been found to contain secondary metabolites in its calyces with biological properties, including organic acids, phenolic compounds, and flavonoids [4,5,6]. *Hs* has been attributed with antihypertensive, hypoglycemic, antioxidant, weight-reducing, and diuretic properties, all of which are associated with its high flavonoid content [7,8,9,10,11,12,13]. These therapeutic effects are particularly relevant given the increasing global prevalence of conditions such as hypertension, diabetes, and obesity in recent years [14,15]. Despite evidence supporting the therapeutic potential of *Hs*, data regarding its possible adverse effects remains limited. Among the toxicological aspects studied, in vivo investigations have reported reproductive toxicity in males secondary to the administration of aqueous extracts. For instance, intrauterine exposure to 250 and 500 mg/Kg of aqueous *Hs* extract during the critical period of sexual differentiation has been shown to reduce sperm count and increase offspring weight at weaning [16]; similarly, in rats, doses of 200 mg/Kg of aqueous extracts obtained via cold and hot extraction caused sperm morphological abnormalities in 43.5% and 52.5% of cases, respectively [17]. It has also been reported that administering *Hs* in drinking water for 12 weeks at a dose of 4600 mg/Kg led to reduced sperm counts in the epididymis and sperm disintegration, while doses of 1150 and 2300 mg/Kg caused epithelial changes and testicular hyperplasia, respectively [18]. In addition, hepatotoxic effects were reported for hydroalcoholic extracts of *Hs* administered 10 to 15 times daily at a dose of 250 mg/Kg in a murine model [19]. Nephrotoxic effects have also been documented at doses ranging from 125 to 2000 mg/Kg, including increased electrolyte excretion and enhanced diuretic activity within the first 18 h [20]. Furthermore, acute exposure to doses between 125 and 2000 mg/Kg has been associated with elevated levels of total protein, albumin, and platelets, while chronic exposure to 5000 mg/Kg resulted in reduced liver weight. Subchronic toxicity was linked to increases in globulin, urea, and creatinine levels, as well as leukopenia and thrombocytopenia [20,21].

However, no evidence has been found regarding the toxicological evaluation of *Hs* in relation to the integrity of genetic material, an essential aspect of comprehensive toxicological assessment. Genetic damage can lead to teratogenic effects and mutations that may result in cancer in exposed organisms and their offspring. Therefore, completing the toxicological profile of *Hs* is particularly important given its widespread use in traditional medicine and as a dietary component. Although toxicological reports for *Hs* are scarce, there is still the possibility of some toxicological risk associated with the high concentration of flavonoids, anthocyanins, and organic acids which may present pro-oxidant effects under specific conditions. For this reason, the present study focuses on assessing the potential genotoxic and cytotoxic effects of the hydroalcoholic extract of *Hibiscus sabdariffa* (EH*Hs*) using the mammalian micronucleus assay recommended by the Organization for Economic Cooperation and Development (OECD), along with quantification of malondialdehyde (MDA) as a biomarker of lipid peroxidation.

## 2. Results

### 2.1. Mothers

#### 2.1.1. Genotoxic and Cytotoxic Evaluation of the EH*Hs* with the Micronucleus Test in Peripheral Blood in Wistar Rats

Figure 1 shows the results of the genotoxic and cytotoxic damage assessment using the micronucleus assay in peripheral blood of Wistar rats exposed to the EH*Hs* during gestation.

Regarding the proportion of polychromatic erythrocytes (PCEs), both the positive control exposed to cyclophosphamide (CP) group and the 2000 mg/Kg dose of the EH*Hs* showed a statistically significant decrease at 120 h *(p =* 0.0001) compared to baseline (0 h), the CP group decreased the number of PCEs from 33 ± 6.44 to 22.2 ± 3.11 and the 2000 mg/Kg dose of the EH*Hs* from 31.8 ± 7.75 to 28.6 ± 11.76, indicating that the CP and the 2000 mg/Kg dose induce bone marrow myelosuppression (Figure 1a).

An increase in the proportion of micronucleated polychromatic erythrocytes (MNPCEs) was observed only in the CP group at 96 and 120 h *(p =* 0.002 and 0.0001, respectively) compared to baseline; this increased from 1.20 ± 0.83 to 4.4 ± 1.51 at 96 h and to 5.4 ± 2.19 at 120 h, indicating that CP induces genotoxicity (Figure 1b).

#### 2.1.2. Cytotoxic Evaluation of the EH*Hs* Through the Quantification of MDA in Mothers

Figure 2 shows the results of MDA levels in the study groups in the different tissues analyzed.

The Figure 2a shows the results of serum MDA quantification in the mothers exposed to the EH*Hs*, as well as in the control groups. As observed, both the CP group and the 1000 mg/Kg dose of the extract showed a tendency toward increased MDA levels compared to the negative control with medians of 0.27, 0.31, and 0.19 nmol/mL MDA, respectively, although this increase was not statistically significant. In contrast, the 500 and 2000 mg/Kg doses of the extract showed a trend toward decreased MDA levels relative to the negative control with medians of 0.023, 0.054, and 0.19 nmol/mL MDA, respectively, but again, the differences were not statistically significant. When comparing MDA medians between the CP group (0.27 nmol/mL MDA) and the groups treated with different doses of the extract, the 500 and 2000 mg/Kg doses (0.023 and 0.054 nmol/mL MDA, respectively) showed significantly lower MDA levels (*p =* 0.011 and 0.007*,* respectively). Likewise, comparisons among the different extract doses revealed that the 500 and 2000 mg/Kg groups had significantly lower MDA levels than the 1000 mg/Kg group (0.313 nmol MDA) (*p =* 0.037 and 0.025, respectively).

At hepatic and renal levels (Figure 2b,c) the study groups showed a very similar pattern. The CP group exhibited a tendency toward decreased MDA levels in both liver and kidney tissues compared to the negative control, although the difference was not statistically significant. In the liver, there was a decrement from 49.34 to 20.68 nmol/g MDA and in the kidney from 51.04 to 35.64 nmol/g MDA. The 500 and 2000 mg/Kg doses of EH*Hs* increased MDA levels in the liver and kidneys relative to the negative control with medians of 352.3, 1014, 49.34 nmol/gMDA, respectively, in the liver 238.9, 401.7, and 51.04 nmol/g MDA, respectively in the kidney, being statistically significant only at the 2000 mg/Kg dose (*p =* 0.032 in the liver and 0.006 in the kidney). In the liver, the 500 and 2000 mg/Kg doses showed significantly higher MDA levels compared to both the CP and 1000 mg/Kg groups (20.68 and 31.49 nmol/gMDA, respectively). In the kidney, the 2000 mg/Kg dose showed significantly higher MDA levels compared to the CP and 1000 mg/Kg groups (35.64 y 59.35 nmol/gMDA, respectively), while the 500 mg/Kg dose showed significantly higher levels compared to the CP group.

### 2.2. Newborns

#### 2.2.1. Morphometric Parameters in Newborns In Utero Exposed to the EH*Hs*

The average number of neonates per group was 12.2 ± 2.16 (negative control), 7.6 ± 2.07 (CP), 10.2 ± 4.02 (2000 mg/kg EH*Hs*), 11.6 ± 2.7 (1000 mg/kg EH*Hs*), and 9.2 ± 1.78 (500 mg/kg EH*Hs*). The number of mated females was 25; we obtained a pregnancy rate of 100%, with a fertility index of 100%; similarly, the death of any of the newborns of the evaluated groups was not presented, obtaining a total of 254 born-alive newborns.

Six newborns were randomly selected from each rat; the sample was defined according to the OECD micronucleus protocol in mammals [22]. The results of the morphometric measurements of newborns in utero exposed to different doses of the EH*Hs*, as well as to the CP and tridistilled H_2_O, are presented in Table 1.

Newborns exposed in utero to the CP showed the lowest weight, height, and abdominal length. Those exposed to the 500 mg/Kg dose of the extract had the highest weight and Lee index (LI), while newborns from the 1000 mg/Kg group exhibited the greatest height and abdominal length but the lowest LI.

#### 2.2.2. Genotoxic and Cytotoxic Evaluation in Newborns In Utero Exposed to EH*Hs*

The results of the genotoxic evaluation in the newborns of rats exposed to the different treatments are shown in Figure 3.

Newborns in utero exposed to CP exhibited the highest proportion of MNPCEs and micronucleated erythrocytes (MNEs) as well as the lowest proportion of PCEs. In contrast, the negative control group showed the lowest proportion of MNPCEs and MNEs (Figure 3).

The Figure 3a presents the behavior of the different groups regarding the proportion of PCEs. As shown, the group exposed to CP displayed a significant decrease (*p =* 0.014) compared to the control group, indicating bone marrow myelosuppression due to CP exposure. Conversely, the groups treated with 500 and 1000 mg/Kg of EH*Hs* showed a significant increase in this proportion compared to the negative control group (*p =* 0.001 and 0.0001, respectively), suggesting that EH*Hs* stimulated erythropoiesis at those doses.

Regarding the proportion of MNPCEs (Figure 3b), both the CP group and all three evaluated doses of EH*Hs* caused a significant increase in this parameter compared to the control group (*p ≤* 0.001), with the CP group showing the most pronounced effect. This indicates that in utero exposure of neonates to CP and EH*Hs* induces short-term genotoxic damage.

Similarly, in utero exposure to CP and the 500 mg/Kg dose of EH*Hs* resulted in long-term genotoxic damage (Figure 3c), as evidenced by a statistically significant increase in the proportion of MNEs compared to the control group (*p =* 0.0001 and 0.014, respectively).

#### 2.2.3. MDA as a Marker of Cytotoxicity in Newborns In Utero Exposed to EH*Hs* and CP

Figure 4 presents the results of MDA levels quantification in kidney, liver, brain, and muscle of newborns in utero exposed to the different compounds.

As shown, the different experimental groups showed a very similar pattern in MDA levels in the tissues. Newborns in utero exposed to CP and to the 500 and 2000 mg/Kg doses of the EH*Hs* tended to show increased MDA levels compared to the negative control group in all the organs analyzed. The greatest increases were observed with the 2000 mg/Kg dose in the liver, kidneys, brain and with the 500 mg/Kg dose in muscle tissue. The 1000 mg/Kg dose exhibited a pattern very similar to that of the control group across the evaluated organs.

## 3. Discussion

According to the World Health Organization (WHO), approximately 80% of the global population continues to use herbal remedies and other elements of traditional medicine as a first line of treatment for various illnesses [2]. Mexico is one of the leading consumers of herbal remedies, with a great diversity of native and introduced plant species used for therapeutic purposes [23,24].

It is commonly believed that plants, being of natural origin, are completely harmless and effective across a wide range of doses. However, for most plants used as herbal remedies, specific safe dosage ranges have not been clearly established, and there is a scarcity of studies evaluating their potential adverse effects especially when compared to the volume of research focused on their therapeutic properties [25,26]. In the case of *Hs*, only seven reports were found evaluating the toxicity of its calyces, six of which focused on the aqueous extract and only one on the hydroalcoholic extract [16,17,18,19,20,21,27], despite the fact that hydroalcoholic extraction yields a higher concentration of flavonoids and phenolic compounds such as quercetin, catechin, and rutin, due to alcohol’s ability to extract a greater number of polar or semi-polar molecules [28]. For this reason, continued evaluation of hydroalcoholic extracts of *Hs* is considered necessary.

Genotoxicity and teratogenicity are mechanistically independent yet closely related processes, as genetic damage, particularly in germ cells or during early organogenesis, can compromise genomic integrity and trigger apoptosis or defective repair. Such alterations impede the proliferation, migration, and differentiation of cells essential for morphogenesis, leading to teratogenic outcomes ranging from minor anomalies to severe malformations or embryonic loss. Therefore, preserving the stability and proper functioning of the genome is critical to ensuring normal embryonic and fetal development [29].

Our results demonstrated that exposure to CP induced myelosuppression, as evidenced by the decrease in the proportion of PCEs. In addition, short-term genotoxic effects were observed, reflected by an increase in the proportion of MNPCEs in both the mothers and their newborns, as well as long-term genotoxic effects, indicated by an increase in the proportion of MNEs in the newborns [30,31,32,33].

The administration of EH*Hs* during gestation in Wistar rats did not induce genotoxicity; however, the 2000 mg/Kg dose did cause cytotoxicity at 120 h, as evidenced by a reduction in the number of PCEs. This decrease is associated with a myelosuppressive effect on the bone marrow.

Newborns from mothers exposed to EH*Hs* exhibited short-term genotoxicity, reflected by an increase in the proportion of MNPCEs at all three evaluated doses (500, 1000, and 2000 mg/Kg), and long-term genotoxicity at the 500 mg/Kg dose. This response does not follow a typical dose-dependent pattern, suggesting the possible presence of a non-monotonic dose–response relationship. This behavior may be attributed to the fact that newborns still possess immature xenobiotic elimination systems, and hepatic enzyme activity begins only days after birth, which delays the clearance of foreign compounds [34,35].

The fact that EH*Hs* exhibited a myelostimulant effect despite existing evidence of their cytotoxicity supports the current understanding of the nature of hematopoietic stem cells, which are believed to possess a unique mechanism of self-renewal and differentiation [36].

The genotoxic effect of *Hs* may be related to its quercetin content, as demonstrated by a bacterial reverse mutation (AMES) test, in which quercetin was identified as responsible for the genotoxic effect of *Hs* [37].

The aqueous extract of *Hs* has been associated with an increase in red blood cell count in healthy adult males [38]. Furthermore, the intake of ascorbic acid and iron obtained from *Hs* calyces has been linked to increased erythropoiesis in patients with hemolytic anemia [39].

In previous reports, it has been shown that doses ranging from 200 to 4600 mg/Kg of aqueous *Hs* extracts have been reported to exert cytotoxic effects on sperm and their germ cell lines [16,17,18]. However, there is also evidence supporting the genoprotective role of *Hs* against clastogenic agents such as CP and 1-nitropyrene (1-NP) [40,41]. These previous studies report that *Hs* has genoprotective effects against genotoxic agents; however, in the present study it was found that at the doses evaluated (500, 1000, and 2000) the EH*Hs* can be genotoxic by itself; although these results seem contradictory, in reality they are complementary, since they provide information about the safety profile of *Hs* on genetic material at different physiological stages and scenarios, although no studies were found evaluating the teratogenic effect of *Hs* for comparison.

Exposure during pregnancy to physical, chemical, biological, or environmental agents may result in birth defects, congenital malformations, growth retardation, fetal death, or even postnatal mortality. Exposure to CP during gestation reduced the average number of newborns per rat to 7.6 ± 2.07 compared to the control group (12.2 ± 2.16 newborns). Moreover, the newborns showed the lowest weight, height, and abdominal length, consistent with previously reported teratogenic effects, including musculoskeletal malformations, and low birth weight and height [30,31,32,33].

There is evidence that intrauterine exposure to *Hs* at a dose of 500 mg/Kg increases the birth weight of newborns [16]; the greater length has also been associated with the growth-promoting effect of the flavonoids present in *Hs* [42].

Newborns from the different evaluated groups showed LI values > 0.31. The LI is used as a tool to estimate adiposity levels in murine models during adulthood; additionally, it allows us to assess the potential metabolic effects of exposure to xenobiotics, as well as to detect developmental alterations. However, there are no established values for adiposity at birth in newborns, so this value may differ due to the unique body proportions of newborns and the amount of brown adipose tissue they possess [43,44].

MDA is a byproduct formed during the oxidation of membrane phospholipids. In the process of lipid peroxidation, reactive oxygen species (ROS) such as hydroxyl (·OH) and superoxide (O_2_·^−^) radicals abstract hydrogen atoms from polyunsaturated fatty acids, generating lipid peroxyl radicals. These radicals undergo cyclization and propagate chain reactions, leading to the formation of cyclic intermediates (endoperoxides), which subsequently degrade into MDA. Due to its dialdehyde structure, MDA is highly reactive and capable of damaging multiple cellular components, such as the genetic material; ROS can form covalent DNA adducts, induce single and double-strand breaks, modify nitrogenous bases (for example, through the formation of 8-oxoguanine), promote depurination or depyrimidination, and disrupt the integrity of phosphodiester bonds. All of these processes can compromise the fidelity of DNA replication and transcription, increase the likelihood of mutations, and activate pathways related to apoptosis or DNA repair. Collectively, these mechanisms establish oxidative stress as a key contributor to genotoxicity and to the onset of various pathologies, including cancer and degenerative or metabolic diseases [45,46,47].

Although the assay used for MDA quantification has certain limitations, such as cross-reactivity with other aldehydes and carbonyl compounds, MDA remains a widely accepted biomarker for assessing lipid peroxidation. Moreover, it is a commonly employed method, which facilitates comparison of our results with previous literature. Finally, to minimize bias, the appropriate control groups were included [46,47].

Regarding the toxicity of EH*Hs*, assessed through the quantification of MDA levels in both exposed mothers and their newborns, the 1000 mg/Kg dose of EH*Hs* administered to pregnant rats exhibited a pro-oxidant effect at the serum level, as indicated by elevated MDA levels compared to the control group, with trends toward a decrease at the 500 and 2000 mg/Kg doses. Although well-established antioxidant mechanisms of EH*Hs* have been recognized, there is also in vitro evidence that molecules such as flavonoids can exhibit pro-oxidant behavior, thus increasing oxidative stress under certain conditions, such as alkaline pH and the presence of transition metals such as Fe and Cu, among others. This could be the case for the observed behavior, which at a dose of 1000 mg/Kg shows a pro-oxidant effect [48,49,50]. At very high doses of the EH*Hs*, almost all flavonoids bind to albumin, leaving very little “free fraction” available for oxidation; this explains why MDA levels decrease in serum at 2000 mg/Kg, whereas at 1000 mg/Kg, the binding sites are not yet saturated, resulting in more “free” compounds and thus increased serum oxidation [51].

In contrast, at the liver and kidney level, the 500 and 2000 mg/Kg doses of EH*Hs* administered to pregnant rats and their newborns also showed higher MDA levels compared to the control group. This may be associated with the hormetic behavior exhibited by some xenobiotics, in which toxic effects are heightened at low doses, desensitization occurs at intermediate levels, and harmful effects reappear at higher doses; this behavior could be associated with a “U”-shaped response curve [52].

The absence of genotoxic effects in mothers, despite increased MDA levels, suggests that the oxidative stress induced by the EH*Hs* was sufficient to alter lipid structures in metabolically active tissues such as the liver and kidneys, but did not reach a threshold capable of inducing chromosomal breaks or bone marrow dysfunction. It is therefore likely that the distribution and metabolism of EH*Hs* led to a localized oxidative stress, without significant or sustained exposure of the DNA.

Evaluating the cytotoxicity and genotoxicity of medicinal plants such as *Hs* calyces during pregnancy is of utmost importance, as pregnancy involves changes in hepatic and renal metabolism, as well as modifications in the cardiovascular system, which increase the susceptibility to toxicological damage caused by the improper use of medicinal plants [53,54,55].

Using the standard body surface area conversion proposed by the Food and Drug Administration (FDA), the tested doses were 500, 100, and 2000 mg/kg, equivalent to approximately 81, 162, and 324 mg/kg in humans, or approximately 4.85, 9.7, and 19.4 g/day for a 60 kg person. Typical consumption of *Hs* infusions varies from 1 to 10 g/day, depending on the preparation. Therefore, according to the results of this study, during pregnancy, doses of ≥ 4.85 g/day should not be consumed due to their genotoxic and cytotoxic impact on the newborn after transplacental exposure; similarly, maternal intake of 19.4 g/day throughout gestation may lead to cytotoxic effects in the mother [56].

Although well-defined safety thresholds for medicinal plant use during pregnancy are lacking, it is essential to conduct studies assessing both acute and chronic exposure and their impact on fetal development to establish a reliable safety profile. In our preclinical model, administration of the EH*Hs* led to elevated markers of cytotoxicity in the mothers and both cytotoxic and genotoxic effects in the offspring following transplacental exposure, underscoring a clear risk profile. Consequently, despite the well-documented antihypertensive, antioxidant, and lipid-improving benefits of *Hs* anthocyanins and phenolics in humans, it is crucial to define the toxicological margin. Given the potential for diuretic and metabolic interactions and the absence of data on placental biodistribution, we recommend advancing to Phase I/II clinical trials in healthy women and low-risk pregnant subjects to characterize tolerability, pharmacokinetics, placental bioavailability, and to establish intake limits that maximize benefit without compromising maternal–fetal safety.

The findings of this study provide valuable insight into the current knowledge gap regarding the safety profile of *Hs* during critical physiological stages such as gestation and fetal development, which could help guide future evaluations in human populations. Although the administered doses were adjusted for animal models, their extrapolation using allometric scaling suggests that similar concentrations may be reached in traditional medicine settings or through the use of non-standardized herbal products. Moreover, they highlight the complexity of natural compound effects, which can vary considerably depending on dose, developmental stage, and cell type. Nonetheless, several limitations of our study warrant acknowledgment. First, we did not assess additional molecular markers of antioxidant activity, such as superoxide dismutase, catalase, and related enzymes which could have provided a more comprehensive view of oxidative defense. Second, to avoid potential pseudo-replication, future investigations should treat each litter as the experimental unit or employ statistical models that incorporate litter effects. Third, our evaluation of neonates exposed in utero to *Hs* was limited to genotoxic outcomes at birth; we did not perform long-term follow-up to determine whether these early effects persist or evolve later in development. Together, these limitations underscore the need for continued toxicological studies of *H. sabdariffa* to establish safety thresholds and inform recommendations for its use during pregnancy.

## 4. Materials and Methods

### 4.1. Hibiscus Sabdariffa Hydroalcoholic Extract

#### 4.1.1. Preparation

The EH*Hs* was prepared according to the methodology proposed by Rodriguez-Saona & Wrolstad (2001) [57]. The calyxes of the Rosalíz variety of the plant were purchased from the commercial establishment “Sam’s Club^®^” (Zacatecas, México); on the packaging the company declares that the plant was sourced under rigorous quality control and traceability procedures and that no pesticides were used. The packaging bears the NOM-051-SCFI/SSA1-2010 compliance seal, which guarantees the accuracy of information regarding origin, batch number, and handling conditions. This study did not include botanical authentication by a specialist. For future studies, we recommend conducting a thorough morphological comparison. The *Hs* calyxes were dried in an oven at less than 50 °C and mechanically pulverized until we obtained a fine powder, which was passed through a sieve to yield particles~250 nm (mesh No. 60). Once we obtained the dry powder, the extract was prepared using a solvent mixture at a 1:4 proportion. The solvent mixture contained distilled water, ethanol, and acetic acid in proportions of 79.3%, 20%, and 0.7% (*v*/*v*), respectively. We bubbled the sample suspension with nitrogen gas to remove dissolved oxygen then sealed in a flask; subsequently, we macerated the plant material under mechanical agitation during 24 h at room temperature, protected from light. Every 24 h, for three times, the macerate was filtered. We stored the supernatant under refrigeration (2–4 °C) and shielded from light, while the residual plant material was returned to the flask and fresh solvent was added. After completing the maceration cycles, we combined all of the collected supernatant and filtered through medium porosity filter paper (pore size 8 μm). We measured the pH’s extract using a potentiometer and adjusted to 2.2, then we concentrated it using a rotatory evaporator under reduced pressure of 100 mbar until obtaining a final volume of approximately 20 mL. The resulting concentrate was frozen at −20 °C and then lyophilized; we stored the final extract in an airtight container at −20 °C and protected from light until its use. An image of the EH*Hs* can be found in Appendix A. Finally, we prepared the different doses evaluated: 2000, 1000, and 500 mg/Kg.

The phytochemical analysis revealed the presence of flavonoids, phenolic acids, and organic acids which are available in Appendix A. Details of the phytochemical analysis such as the gas chromatography-mass spectrometry and the identified compounds in the EH*Hs* are available in Appendix A.

#### 4.1.2. Experimental Design

The cytotoxic and genotoxic potential was evaluated in twenty-five clinically healthy 3-month-old pregnant Wistar rats and in six of their newborns. Animals were housed with appropriate environmental enrichment, with a maximum of five animals per cage, and were randomly assigned to the different experimental groups. The rats were maintained in polycarbonate cages at a temperature of 22 ± 2 °C, under a 12/12 h light/dark cycle and a relative humidity of 50 ± 10%. They were fed with standard laboratory pellet food from Purina^®^ brand and purified water ad libitum.

The sample size in this study was determined based on the guidelines established in the protocol for the micronucleus assay in mammals, which specifies that each experimental group should consist of a minimum of 5 animals. Although previous genotoxicity studies commonly include six or more animals per group, the present study deliberately employed the minimum required number in order to adhere to the Reduction principle of the 3Rs (Replacement, Reduction, and Refinement). This decision aligns with the recommendations of the OECD, which advocate for minimizing the use of animals in scientific research without compromising the validity and reliability of the results [30,58]. We used distilled water at a proportion of 0.1 mL/10 g of weight as the negative control to establish a basal line and cyclophosphamide at 30 mg/Kg dose administered only the last 2 gestational days as positive control since this was the dose defined as clastogenic by the OECD. CP must be bioactivated to exert its toxic effects; this process is mediated by microsomal monooxygenases of the cytochrome P-450 system, which convert CP into 4-hydroxycyclophosphamide (4OHCP). 4OHCP is rapidly converted into phosphoramide mustard (PM) and acrolein (AC). PM is the metabolite responsible for CP’s mutagenic effect, and some studies suggest that both PM and AC contribute to its teratogenic effects.

The doses of the EH*Hs* administered in this study were determined based on the protocol for the micronucleus assay in mammals, which states that, in the absence of prior genotoxicity data for the test substance, the highest dose used should be 2000 mg/Kg. Subsequent doses should be established by reducing the previous one by 50%, in a stepwise manner [30].

The EH*Hs* and the controls were administered orally with an esophageal cannula as indicated in Table 2.

The different compounds were administered only during the last 5 days of gestation because, after exposure to EH*Hs*, any chromosomal damage in erythrocytes becomes visible in the PCEs that appear in peripheral blood approximately 24–48 h after the injury and stabilizes in circulating erythrocytes at 72 h. Sampling during the last 5 days ensures that sufficient time has elapsed between the first day of administration and the final sample for micronuclei to form and be detected [30,59,60]. We acknowledge that limiting exposure to the last 5 days of gestation may not capture earlier developmental effects, and this remains a limitation of our experimental design.

This study was approved by the Research Ethics Committee from the Autonomous University of Zacatecas’s nursing school under the registration number: CI-UAE-02-2024.

#### 4.1.3. Gestation Induction

Daily vaginal cytological analyses were carried out on females to confirm the estrus phase in the reproductive cycle; subsequently, a male was mated with three females for 24 h, then, a vaginal lavage was performed with 0.1 mL of sterile water using a micropipette equipped with a pre-rounded tip. The resulting solution was placed on a pre-labeled slide; then, it was observed in an optical microscope at 10× to detect the presence of sperm, which indicated the onset of gestation.

Once the female’s pregnancy was confirmed, the gestation period was scheduled and the administration of the EH*Hs* and control groups was arranged.

#### 4.1.4. Sample Collection

Blood samples were collected from pregnant rats; this was carried out through a small excoriation at the tip of the tail. Blood smears were prepared in duplicate for each rat before the first dose was administered (baseline sample) and then every 24 h for six days (0, 24, 48, 72, 96, and 120) [61]. Once the gestational period was complete, six newborns per rat were randomly selected, maintaining a 1:1 sex ratio according to what was previously established by Gomez-Meda et al. (2004) [62]. Blood samples were collected from the tail of the newborns the day they were born and prepared blood smears by duplicate. Mothers and their newborns were sacrificed on the delivery day.

#### 4.1.5. Sacrifice

The animals were sacrificed by placing them in an ether chamber until they were fully anesthetized, after which they were exsanguinated via cardiac puncture and a blood sample was obtained from the ventricular cavity of mothers; the newborns were placed in the same chamber until they exhibited no response to stimuli; we also observed the cessation of respiratory movements, as well as the lack of response to neurological reflexes to confirm the death of the newborns. Blood was centrifuged and serum samples were stored at −20 °C until analysis. Liver and kidney from pregnant rats and liver, kidney, brain, and skeletal muscle from neonates were obtained by dissection, tissues were rinsed with phosphate buffer solution at pH 7.0 for subsequent homogenization. The phosphate buffer was used exclusively to rinse the tissues prior to homogenization, with the aim of removing blood residues and minimizing interference in the quantification of malondialdehyde. Since the washing did not involve cell lysis or prolonged exposure to the environment, we did not add protease inhibitors.

#### 4.1.6. Morphometric Parameters

In the newborns of the exposed rats, body weight was determined through a digital mini-balance (Jinwen); nasoanal length and abdominal length were determined using a digital vernier caliper (Mitutoyo). LI was calculated based on the methodology proposed by Lee in 1928 [63], by the quotient of the cube root of body weight (g) and nasoanal length (mm).

#### 4.1.7. Determination of Genotoxic and Cytotoxic Damage by Micronucleus Assay in Pregnant Rats and Their Newborns

All blood smears from both mothers and newborns were air-dried and fixed with absolute ethanol for 10 min before being stained with acridine orange. Manual analysis was performed using an Olympus CX31 microscope equipped with epifluorescence and a 100X oil immersion objective.

In newborns, the number of MNEs was determined in a total of 10,000 total erythrocytes (total erythrocytes: polychromatic and normochromatic erythrocytes). The number of MNPCEs was counted in 1000 PCEs, and the PCEs counted in 1000 TE. In adult rats, only the proportion of PCMNEs and PCEs was counted [64,65].

The number of MNEs was not counted in mothers; this is because the spleen in adult mammals is fully functional and can easily delete the damaged erythrocytes [30,65,66].

#### 4.1.8. Quantification of MDA in Different Organs as a Marker of Cytotoxicity

Quantification of MDA was performed on serum, liver, and kidney samples from the mothers and liver, kidney, muscle, and brain samples from their newborns using the modified method proposed by Uchiyama and Mihara (1978) [67].

A 10% tissue homogenate was prepared with 1.15% KCl and a 1/12 N-sulfuric acid homogenate of blood serum was prepared. A sample of 0.5 mL was taken and transferred to a previously labeled tube to which 3 mL of 1% phosphoric acid and 0.3 mL of 0.6% thiobarbituric acid were added. The mixture was heated in a boiling water bath for 45 min. After cooling, 3 mL of 1-butanol was added. The mixture was shaken by inversion for 20 min and then centrifuged at 3000 rpm for 10 min. The organic phase was read using a Jenway UV/Vis 6715 spectrophotometer (Stone, United Kingdom) at a wavelength of 534 nm. The level of MDA in the samples was determined using a calibration curve with 1,1,3,3-tetramethoxypropane. The calibration curve for MDA was constructed using concentrations ranging from 0 to 16 nmol; also, the calibration curve was linear within this range, with a correlation coefficient (R^2^) of 0.998. (Sigma Chemical, St. Louis, MO, USA) (Uchiyama et al., 1978 [67]).

#### 4.1.9. Statistical Analysis

The Shapiro–Wilk and Kolmogorov–Smirnov test were used to establish that the data were normally distributed.

For the frequencies of PCEs, MNPCEs, and MNEs, the results obtained were expressed as mean ± standard deviation per group. For rats, comparisons were made between each group and its respective baseline value (0 h), using the analysis of variance (ANOVA) for repeated measures and the Bonferroni adjustment test for multiple post hoc comparisons.

For newborn rats, the litter was used as the experimental unit (*n* = 6/group), and pups were the unit of observation and analysis. Intergroup comparisons were made using the Kruskall–Wallis analysis with Dunn’s post hoc.

Data for MDA concentrations were expressed as a median with the maximum and minimum. Intergroup comparisons were made using the Kruskall–Wallis analysis with Dunn’s post hoc.

All statistical analyses were conducted using SPSS software (version 25) for Windows^®^. A *p*-value of ≤0.05 was considered statistically significant.

## 5. Conclusions

The EH*Hs* induced myelosuppression in pregnant rats and genotoxic damage in their newborns, in addition to increasing lipid peroxidation, as evidenced by elevated MDA levels in the various organs analyzed. While these findings provide valuable information on the safety profile of *Hs* during pregnancy, they should be interpreted with caution, as the study is limited to sampling at a single late-gestation time point. Consequently, further investigations, such as studies across diverse gestational windows and long-term follow-up, are required to fully characterize both maternal and fetal risks. Therefore, the use of *Hs* extracts is recommended only under medical supervision.

## Figures and Tables

**Figure 1 ijms-26-07448-f001:**
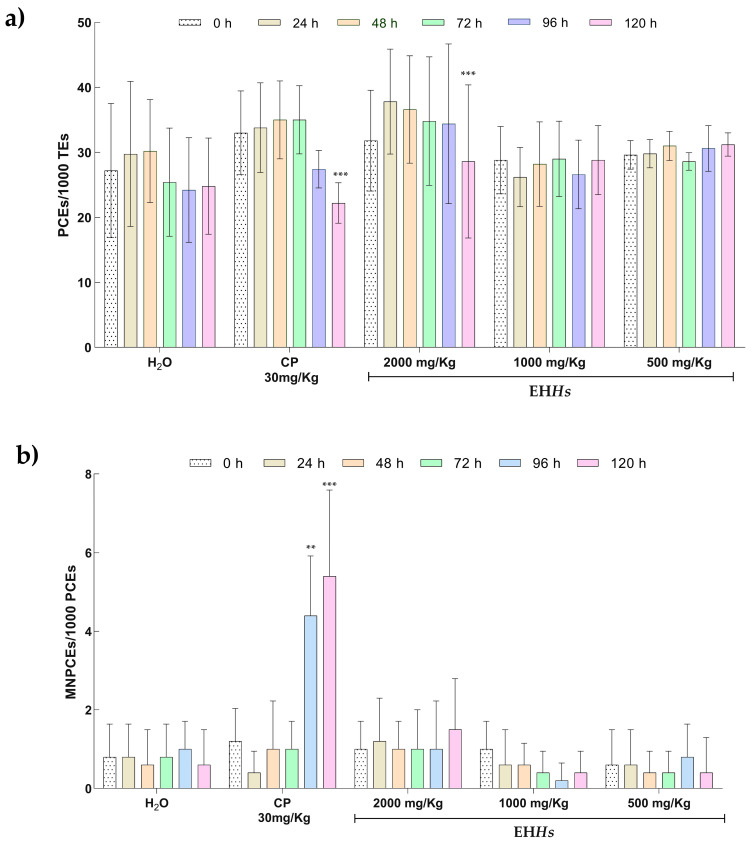
Results of cytotoxic and genotoxic parameters of Wistar rats exposed to EH*Hs*. (**a**) Proportion of polychromatic erythrocyte in the study groups. (**b**) Proportion of micronucleated polychromatic erythrocytes in the study groups. The statistical analysis was performed using a repeated measures ANOVA test with a Bonferroni post hoc. Intragroup comparisons were performed between baseline samples (0 h) against the following sampling times: 24, 48, 72, 96, and 120 h. PCE: Polychromatic erythrocytes; TE: total erythrocytes; MNPCE: micronucleated polychromatic erythrocytes; CP: cyclophosphamide; H_2_O: tridistilled water. ** *p* ≤ 0.01, *** *p* ≤ 0.001.

**Figure 2 ijms-26-07448-f002:**
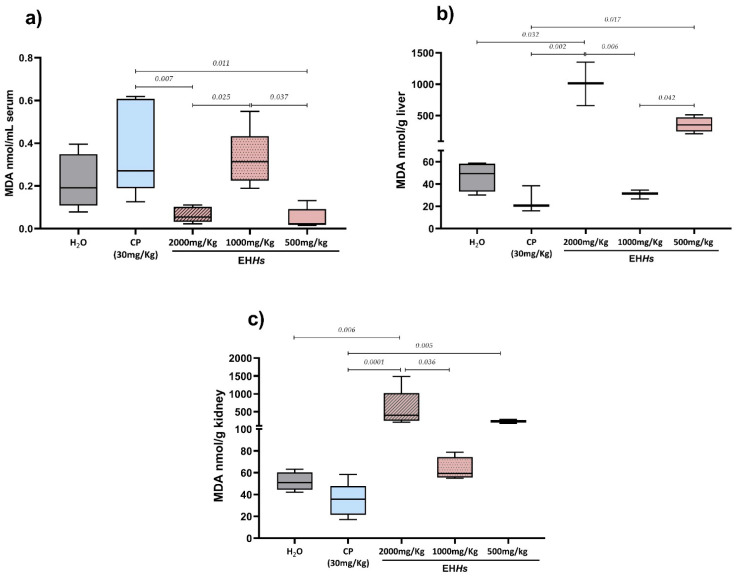
Malondialdehyde (MDA) levels in tissues in the different study groups: (**a**) MDA concentrations in serum, (**b**) MDA concentrations in liver, (**c**) MDA concentrations in kidney. The statistical analysis performed was Kruskal–Wallis with a Dunn–Bonferroni post hoc. MDA: malondialdehyde; nmol: nanomole; mL: milliliter; g: gram; CP: cyclophosphamide; H_2_O: tridistilled water.

**Figure 3 ijms-26-07448-f003:**
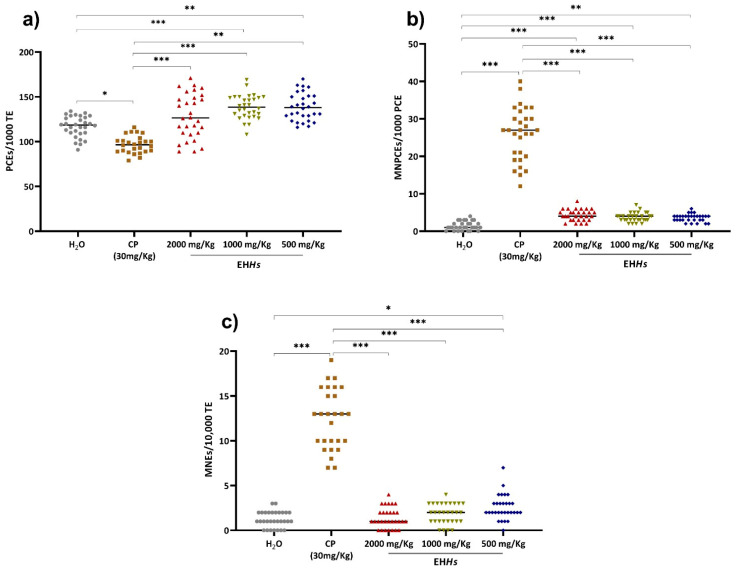
Proportion of PCEs, MNPCEs, and MNEs in peripheral blood of newborns in the study groups. (**a**) Proportion of polychromatic erythrocyte, (**b**) proportion of micronucleated polychromatic erythrocytes, (**c**) proportion of micronucleated erythrocytes. The results are expressed as the median (minimum–maximum). The statistical analysis performed was a Kruskal–Wallis test with a Dunn–Bonferroni post hoc. H_2_O: water; CP: cyclophosphamide; PCE: Polychromatic erythrocytes; TE: total erythrocytes; MNPCE: micronucleated polychromatic erythrocytes; MNE: micronucleated erythrocytes; *: *p* ≤ 0.05; **: *p* ≤ 0.001; ***: *p* ≤ 0.0001.

**Figure 4 ijms-26-07448-f004:**
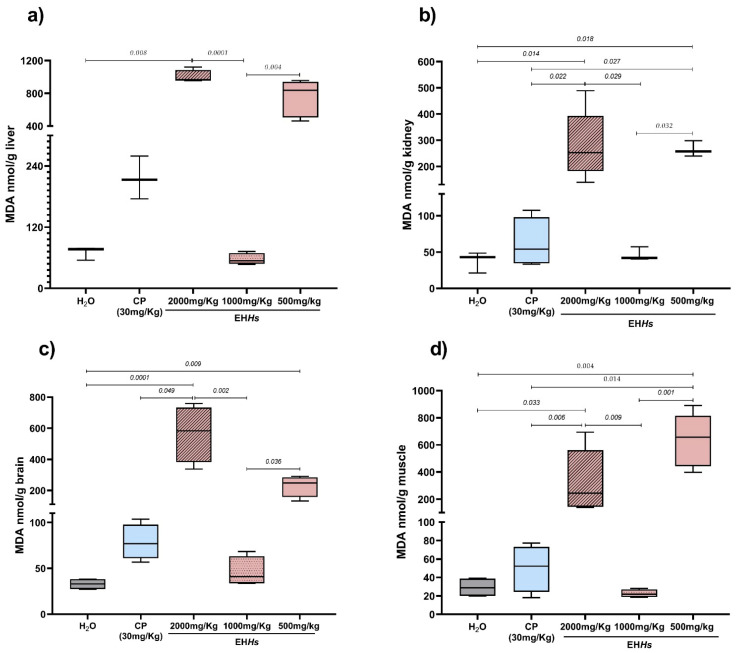
Results of the quantification of malondialdehyde (MDA) in different tissues of newborns in utero exposed in the study groups. The statistical analysis performed was Kruskal–Wallis with a Dunn–Bonferroni post hoc. MDA: malondialdehyde, nmol: nanomole, mL: milliliter, g: gram; (**a**) MDA concentrations in liver from newborns in utero exposed to EH*Hs*. (**b**) MDA concentrations in kidney from newborns in utero exposed to EH*Hs*. (**c**) MDA concentrations in brain from newborns in utero exposed to EH*Hs*. (**d**) MDA concentrations in muscle from newborns in utero exposed to EH*Hs*.

**Table 1 ijms-26-07448-t001:** Morphometric parameters in newborns exposed in utero to EH*Hs*.

Groups of Study	H_2_O	CP 30 mg/Kg	EH*Hs* 500 mg/Kg	EH*Hs* 1000 mg/Kg	EH*Hs* 2000 mg/Kg
Weight (g)	6.20 (4.6–6.9)	4.95 (3.8–5.8)	6.45 (4.0–7.1)	5.85 (5.25–7.10)	6.35 (4.22–7.40)
*p*-value	b ***	d ***			b ***
Height (mm)	44.15 (37.80–50.40)	34.90 (30.50–40.20)	38.30 (31.90–45.60)	44.20 (39.30–48.60)	42.60 (38.40–53.40)
*p*-value	b ***	c *		b ***	b ***
Abdominal length (mm)	13.0 (11.40–14.10)	12.25 (10.20–13.70)	12.60 (10.20–13.90)	13.10 (11.60–15.70)	12.65 (10.30–17.30)
*p*-value		d **			
Lee index	0.5154 (0.48–0.54)	0.5223 (0.49–0.55)	0.5404 (0.50–0.56)	0.5153 (0.49–0.55)	0.5243 (0.43–0.56)
*p*-value	c ***	c *	d ***		c *

The results are expressed as the median (minimum–maximum). The statistical analysis performed was a Kruskal–Wallis test with a Dunn–Bonferroni post hoc. H_2_O: water; CP: cyclophosphamide; g: grams; mm: millimeters. a: H_2_O; b: CP; c: EH*Hs* 500 mg/Kg; d: 1000 mg/Kg; e: EH*Hs* 2000 mg/Kg; *: *p* ≤ 0.05; **: *p* ≤ 0.001; ***: *p* ≤ 0.0001.

**Table 2 ijms-26-07448-t002:** Description of experimental groups in the study.

Group	Treatment	Dose	Administration Route	Administration Time	
1	EH*Hs*	2000 mg/Kg.	Orally	5 days	Each 24 h from gestational day 16 to 20.
2		1000 mg/Kg.			
3		500 mg/Kg.			
4	Tridistilled H_2_O (Negative Control)	0.1 mL/10 g of weight.			
5	Cyclophosphamide (Positive control)	30 mg/Kg.		2 days	Gestational day 19 and 20.

## Data Availability

Data is contained within the article and Appendix A.

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
