# Peer review of "Genomic and Cytotoxic Damage in Wistar Rats and Their Newborns After Transplacental Exposure to *Hibiscus sabdariffa* Hydroalcoholic Extract"

_ijms, 2025, doi:10.3390/ijms26157448_

Round 1
Reviewer 1 Report
Comments and Suggestions for Authors
Dear authors
Thank you for submitting your manuscript. After reviewing it, I believe that your study presents relevant research on the possible genotoxic and cytotoxic effects of Hibiscus sabdariffa extract during pregnancy. Overall, I find your study interesting and given its content, it could be of great interest to the scientific community in this field. However, there are some details that need to be addressed before your research can be considered for its publication. Therefore, below is a list of comments, questions, and detailed suggestions that may help you improve the clarity of your manuscript:
Major comments
P1, line 16. Please consider specifying in the abstract that the extract was derived from dried calyces, as this detail is important in understanding its chemical composition.
P1, line 17. Could you please clarify what "n" refers to in this context? Is it meant to indicate the sample size, or is there a possible typographical error? If it does denote sample size, it might be helpful to specify the exact value (n = X) for clarity. Please explain.
P1, line 32–33. The justification for the use of Hs is very interesting. But I think it could be stronger if you added information or data that describes or justify its use. For example, it may be difficult to find data on the use of this plant in traditional medicine. However, you could include data on the incidence of the diseases for which it is used. This could reflect the impact of this species in traditional medicine.
P2, lines 19–20. Please explain why gestational days 16–20 were selected for administration. The most sensitive period in embryo-fetal development is the first trimester, in rats about 6–15 gestational days. So, why did you administer the treatments at a time when fetuses are most resistant to these alterations? Please explain in detail.
P2, line 43. Please clarify whether the cited toxicological findings in males involved hydroalcoholic or aqueous extracts (or what kind of extract). This distinction is important, as the chemical composition varies between extraction methods.
P2, lines 60–69. It is important to mention the gaps in toxicological information regarding the use of Hs. However, even if there are no toxicological reports or studies of this species, it is advisable to provide data showing what type of toxicity can be expected based on the main metabolites of this species. This will provide a stronger background for your research and support the use of the methodologies employed (if we know what we expect, we will know how to identify it).
P2, lines 77-85. I recommend that the description of your results follows a temporal sequence in which you describe the frequency of PCEs or MNPCEs at time zero, and how it behaves at 24, 48, 72, 96, and 120 h. In this regard, it would be good to also include values (average ± SEM) that allow observation and comparison of the differences between groups.
P2, lines 81, 85. In the text you refer to graphs in panel A and B, but in the referenced graphs the labels appear as lowercase a and b. This can cause some confusion, so I recommend homogenizing this nomenclature.
P3, fig a, b. I suggest that in both graphs, instead of organizing the X-axis with the 5 treatments (5 groups, each including the 6 time points), you organize it by the 6 sampling times (0, 24, 48, 72, 96, 120 h). Then, each bar within each time point would correspond to the 5 treatments (H2O, CP, and the 3 doses of Hs). This would help to more efficiently visualize the differences in frequencies between the groups at each time. Also, please indicate the dose of CP (CP 30 mg/kg), not just CP.
P3, fig a, b. On the other hand, it is striking that in your experiment the frequency of PCEs and MNPCEs with a dose of 30 mg/kg of CP increases up to 96 h. While the literature reports that with 20 mg/kg of CP, a significant increase in both biomarkers can be observed as early as 48 h. Please develop this in the discussion section.
P3, line 93. In the figure caption you indicate statistical differences using ** and *** associated with p-values. However, it is not clear what groups are being compared in the graph. It is more important to state that treatment 1 differed from treatment 2 than to say ** means p < 0.01 or *** means p < 0.001. That is, in the figure legend, you should just indicate p < 0.05 but describe more clearly which comparisons are represented by the asterisks or symbols. For example: statistical differences vs **H2O. This means that all bars with ** are statistically different from the H2O group.
P4, figure 2a, b, c. Please homogenize the nomenclature in the text and figures (either uppercase A, B, C or lowercase a, b, c in both). Also, do not just write CP, but CP 30 mg/kg in the graphs. And in the figure legend, describe in more detail the statistical differences between treated groups: define all statistical symbols and comparisons.
P4, lines 103-124. The description of your results shown in figure 2 would be clearer if you included numerical values and not just a description of trends when comparing the effect of the different treatments.
P4, lines 127–129. This section lacks a lot of information for a complete somatometric analysis. Starting with maternal–fetal parameters such as: number of mated females, number of pregnant females, pregnancy index, fertility index, maternal weight gain, females with live or dead newborns, live newborns, live newborns per litter, dead newborns per litter, etc. This would show the effects that the treatments had on gestation and the offspring.
It is well documented that CP administration during rodent gestation produces serious structural malformations, growth retardation, and embryotoxicity. The absence of teratogenic effects in your results is the product of your experimental design (CP was administered during advanced developmental stages that did not allow teratogenic effects to occur). Therefore, you must justify in the methodology or discussion why you decided to omit this valuable information; as you said, there are no reports on the toxic effects of Hs, so the evaluation of its teratogenic effects (alongside the genotoxic ones) would have been of great value.
P5, lines 147–149. In addition to explaining how the negative and positive controls (CP) behaved, it is also necessary to describe the behavior of the evaluated parameters in the three groups treated with Hs.
P5, table 1. The results of your table are interesting, but the way you indicate statistical differences is confusing. These differences should not only be mentioned in the table footnote, but also shown clearly in the table itself. I see you are using letters to denote differences, which is good. However, you are making it more complex than necessary. The description of your differences can be simpler. First, instead of placing the exact p-value in the table, just mention in the table footnote that p < 0.05. Second, place the letters where appropriate in each cell of the table. Then explain what those letters mean. For example, a clear and simple way would be: statistically significant differences (p < 0.05) vs aH2O, bCP 30 mg/kg, cHs 2000 mg/kg, d etc. In this statement, a means that anything with the letter a is statistically different from the H2O group, b from the CP 30 group, c from Hs 2000, and so on. This way of presenting the differences is much simpler and clearer. This would improve transparency and allow readers to assess statistical significance more precisely.
P5, lines 145–150. Before explaining your results, describe how your groups were formed: how many neonates were obtained in each litter, and therefore in each group. And explain why only 30 neonates per group were used (how were they selected?).
Also, you need to explain why in the neonates another parameter (MNEs) was determined that was not evaluated in the pregnant females.
P5, lines 157–161. You report genotoxic effects in neonates at all doses, but these were not observed in the mothers. So, this discrepancy warrants deeper discussion. For example, could developmental stage or pharmacokinetics explain the higher susceptibility? What do you think? Please explain this in the discussion section.
P6, fig 3a. I find it striking that they present the data on genotoxicity in newborns in table format. When the same genotoxicity data in mothers was shown as bar graphs. I consider it very important that the structure of your results be consistent. Therefore, I believe it is necessary that the genotoxicity data for newborns be shown in the same format as the genotoxicity data for pregnant females. Not to mention that it is not necessary to show the P values in your table (or graph). As I have been explaining, just indicate the differences with letters or symbols and describe them in the footnote of the table/graph next to the P value <0.05.
P6, fig3b, c, and d. Why are you using box and whisker plots in graphs b, c, and d, instead of bar graphs as you did with these same parameters in mothers? If you want to show the dispersion of the data, it would also be useful to do the same with the data for mothers in Figure 2. To do this, you can use bar charts that, in addition to the mean and standard error, also show the dispersion of the data in the form of dots.
The data shown in graphs 3b, c, and d are the same as those in table 3a. It makes no sense to repeat the same data again in graph format. Especially since table a already shows the minimum and maximum values for each group in parentheses. Ergo, there is no new information that justifies the use of scatter plots that repeat the same data. Please standardize these results: remove Table 3a, and instead of box and whisker plots, use bar graphs, as you did with the genotoxicity data for pregnant females.
P7, fig 4. Please address the same comments I made in figure 2.
P8, lines 212-218. The information in this section is background information that should be explained in more detail in the background section (to understand, based on the bioactivation of CP, how this molecule causes the formation of micronuclei and alterations in embryo-fetal development). In this section, rather than explaining the mechanisms of CP, they should only be briefly revisited to highlight or contextualize some of your findings. Remember that the discussion section focuses on comparing your results with those of other researchers, in order to explain, based on the scientific literature, the reasons for the differences or similarities found.
P8, line 223. You mention that the Hs extract caused myelosuppression, as evidenced by a decrease in PCEs, but you do not mention at what point this occurred, which is extremely important information. Please add this information.
In this context, it is necessary to mention that the statistical differences in the graph are very confusing; the asterisks on the 120-hour bar for Hs 2000 mg/kg do not indicate from whom they are different. And since the averages at each sampling time are similar to those at 120, the question is: compared to whom is the frequency of PCEs at 120 different in the Hs 2000 mg/kg group? I am not sure if there is actually myelosuppression, since the frequency of PCEs at 120 h is very similar to the frequency of PCEs at 0 h (physiological levels). Please explain your findings in detail.
In addition, I believe it is necessary to discuss at length the fact that in your group treated with CP 30 mg/kg, no decreases or increases in the frequencies of PCEs and MNPCEs (respectively) were observed before 96 hours. When the scientific literature reports that at doses of 20 mg/kg, the decrease and increase in the frequency of these markers can be observed after 24 hours, reaching their peak between 36 and 48 hours post-treatment. It should also be noted that several articles report a marked normalization of these markers, leading to the frequency of these biomarkers being practically normal (physiological or basal) between 96 and 120 hours post-treatment.
P8, line 224-128. It is very interesting that no genotoxic effects derived from Hs treatment were observed in pregnant females, while these effects were observed in neonates. Please expand your discussion by explaining what might be causing this phenomenon.
P8, line 229. The long-term genotoxicity observed only at the lowest dose of Hs (500 mg/kg) raises questions. Please discuss the possibility of a hormetic effect or non-monotonic dose-response.
P8, lines 235-245. You mention the molecules and background that may be responsible for the genotoxic and cytotoxic effects found after Hs administration. However, you need to explain the mechanism by which quercetin generates genotoxicity in a murine model and correlate it with your results.
On the other hand, their results show that Hs was myelostimulant, but the studies they cite show that Hs is cytotoxic in other cells. This contradiction raises questions that need to be discussed and clarified.
Finally, they also mention that Hs has reports of genoprotection against CP. This is interesting because it raises questions such as: if in your experiment Hs induced genotoxicity in neonates in the short and long term, how does this finding relate to previous reports that indicate that Hs has protective effects against CP-induced genotoxicity? Please expand your discussion on this important point.
P9, lines 260-270. The explanation of MDA levels and flavonoid saturation is intriguing but needs stronger support. Please consider citing experimental studies with similar conditions to support this mechanistic hypothesis.
P9, lines 273-282. Consider adding a section to your discussion that delves into the experimental strengths and limitations, contextualizes your findings, and directs those interested in continuing this research to the most important aspects that should be explored.
P9, lines 283–303. Your extract preparation protocol is thorough. However, you need to provide more technical details so that anyone interested can repeat your methodology: at what refrigeration temperature was the macerate stored? What were the conditions of temperature, reduced pressure, and time under which the extract was concentrated in the rotary evaporator? Why did you concentrate the extract in a rotary evaporator before freeze-drying, rather than freeze-drying it directly? Is this important in the context of metabolite stability? You should mention whether the chemical stability of the extract was evaluated over time, as this could influence metabolite bioavailability. What were the freeze-drying conditions used? In addition, please cite recent studies that support each procedure used in your methodology. It is also necessary to explain the procedures used to carry out the phytochemical analysis. Consider adding the results of this analysis to the results section, as this will help to better contextualize your findings.
P10, lines 305-335. Specify in your design that the n for each group of females was 5 (n=5). Although OECD 474 (Micronucleus Test) indicates that n=5 is sufficient to demonstrate genotoxic effects, consider that similar studies typically use at least n=6 to ensure the statistical robustness of the results. And that OECD-414 recommends n=15 for teratogenicity tests. Therefore, it is important to justify the use of n=5 in your design; cite research that supports your design.
Please discuss the possible limitation of having n=5 per group, and only six neonates per mother per group. How might this sample size affect the generalizability of your findings?
P10, table 2. Why did you use Tridistilled H2O as a negative control? Although it is not toxic, it is known that the lack of ions in ultrapure water can alter the cellular osmotic balance. While it is true that the OECD does not prohibit its use, some agencies (EMA, FDA) recommend standard vehicles such as purified water or saline. Please explain.
P10 lines 322-326. The presence of sperm only indicates that mating took place, but does not guarantee that the mated females are pregnant. To reduce the margin of error, in addition to the presence of sperm, it is necessary to demonstrate that copulation took place during the fertile phase of the rat's estrous cycle. Therefore, the identification of early estrus, estrus, or late estrus during mating is necessary. Please supplement your information.
P10, 327-328. In this section, you should explain why the compounds were administered at a late stage of development, when alterations are unlikely to occur, rather than during sensitive periods of development, in order to guarantee the absence of teratogenic effects of the extract, as suggested by the OECD. Although your research focuses on genotoxic effects, given your experimental design that includes intrauterine development, it is necessary to clarify these methodological details to ensure the correct use of experimental animals for maximum benefit.
P10, lines 333-336. Explain why only six neonates per litter were selected and what the selection criteria were. Clearly describe the condition of the neonates obtained for the corresponding determinations and when the blood samples were taken. Cite research that supports your design.
P10, lines 338-345. The use of ether as an anesthetic is no longer considered ethical or recommended due to its respiratory irritability, risk of explosion, and prolonged suffering in animals. Exsanguination euthanasia procedures are recommended when cleaning, fixing, and tissue removal for histological analysis purposes are justified. If this is not to be done, there are much simpler procedures in adult rats for ethical euthanasia, such as cervical dislocation, ketamine + xylazine, sodium pentobarbital, etc. For fetuses and neonates, ether as an anesthetic or for euthanasia purposes is also unethical. Instead, pentobarbital, controlled hypothermia, or decapitation (if justified by the methodology) are recommended. To say that a newborn has died, the absence of stimuli is not enough, as it may only be anesthetized. Please describe in detail how you ensured the death of the newborns.
Your methodology mentions that the pregnant females were anesthetized with ether. This is confusing, since your methodology mentions that the pregnancy was carried to term, which means that all the offspring were born and no female should have been pregnant at the time of sacrifice. Please correct or explain what you mean by pregnant females.
P11, 368-362. Why were the same parameters not evaluated in the genotoxicity study of mothers and newborns? I understand that micronucleated erythrocytes (MNE) refer to mature erythrocytes in which long-term genotoxicity was determined in newborns. Therefore, it is necessary to describe them in this way, referring to them as micronucleated normochromic erythrocytes. The term MNE is very general (it is not known whether they are young or mature erythrocytes) and can be confusing for people who are not in the field. Explain why MNE in neonates was determined in 10,000 cells and not in 1,000 as in PCEs. This information is important for understanding your methodology.
P11, lines 380-382. The description of your statistical analysis is very limited; simply stating that X analysis was used for parametric variables and Y analysis for non-parametric variables is very limited. Instead, please provide more information and specify the data or variables analyzed with certain tests. For example, in the micronucleus assay, it is not advisable to use repeated measures analysis of variance (ANOVA) that compares only with the baseline (0 h), as it ignores interactions between time and treatment. It is more robust to use a two-way ANOVA for repeated measures, as they have two factors: treatments and sampling days.
P12, lines 388-393. I recommend softening the conclusion. Instead of discouraging use entirely, consider suggesting that further evaluations are necessary and that caution should be exercised during pregnancy. Also mention the methodological or approach limitations of your study.
Minor comments
- Review and maintain consistency in acronyms such as: EHHs vs EEHs, PCEs vs PCE, etc.
- Carefully review the use of abbreviations, as there are terms that you show in parentheses at the beginning of your manuscript and then show again in parentheses in the last sections of your document.
- Standardize the way you present figures; for example, sometimes you write 2,000 and other times 2000.
- Try to reduce passive voice to improve readability.
- Ensure all references are correctly formatted and up to date.
Thank you for your submission. With clarification and improvements in the points noted above, your manuscript can make a meaningful contribution to the literature on the safety profile of medicinal plant use during pregnancy.
Author Response
Response to Reviewer 1 Comments
Thank you very much for taking the time to review this manuscript. Below are the detailed responses to the comments made. The changes were highlighted in red and the corrected file was resubmitted.
Point-by-point response to Comments and Suggestions for Authors
Comments 1: P1, line 16. Please consider specifying in the abstract that the extract was derived from dried calyces, as this detail is important in understanding its chemical composition
Response 1: The word "dried calyces" is added to line 17 of the summary.
Comments 2: P1, line 17. Could you please clarify what "n" refers to in this context? Is it meant to indicate the sample size, or is there a possible typographical error? If it does denote sample size, it might be helpful to specify the exact value (n = X) for clarity. Please explain
Response 2: It's a typographical error, the correct word is “newborns” (Line 18)
Comments 3: P1, line 32–33. The justification for the use of Hs is very interesting. But I think it could be stronger if you added information or data that describes or justify its use. For example, it may be difficult to find data on the use of this plant in traditional medicine. However, you could include data on the incidence of the diseases for which it is used. This could reflect the impact of this species in traditional medicine
Response 3: The following paragraph is added “These therapeutic effects are particularly relevant given the increasing global prevalence of conditions such as hypertension, diabetes, and obesity in recent years [55]” (lines 42-44)
Comments 4: P2, lines 19–20. Please explain why gestational days 16–20 were selected for administration. The most sensitive period in embryo-fetal development is the first trimester, in rats about 6–15 gestational days. So, why did you administer the treatments at a time when fetuses are most resistant to these alterations? Please explain in detail
Response 4: The different compounds were administered only during the last 5 days of gestation because, after exposure to EHHs, any chromosomal damage in erythrocytes becomes visible in the PCEs that appear in peripheral blood approximately 24–48 h after the injury and stabilizes in circulating erythrocytes at 72 h. Sampling during the last 5 days ensures that sufficient time has elapsed between the first day of administration and the final sample for micronuclei to form and be detected [34,36,61]. (This explanation was added on lines 387-392 of the manuscript)
Comments 5: P2, line 43. Please clarify whether the cited toxicological findings in males involved hydroalcoholic or aqueous extracts (or what kind of extract). This distinction is important, as the chemical composition varies between extraction methods.
Response 5: It was specified that it was "secondary to the administration of aqueous extracts" (Lines 46-47)
Comments 6: P2, lines 60–69. It is important to mention the gaps in toxicological information regarding the use of Hs. However, even if there are no toxicological reports or studies of this species, it is advisable to provide data showing what type of toxicity can be expected based on the main metabolites of this species. This will provide a stronger background for your research and support the use of the methodologies employed (if we know what we expect, we will know how to identify it)
Response 6: The following paragraph has been added: "Although toxicological reports for Hs are scarce, there is still the possibility of some toxicological risk associated with the high concentration of flavonoids, anthocyanins and organic acids which may present prooxidant effects under specific conditions". (Lines 69-72).
Comments 7: P2, lines 77-85. I recommend that the description of your results follows a temporal sequence in which you describe the frequency of PCEs or MNPCEs at time zero, and how it behaves at 24, 48, 72, 96, and 120 h. In this regard, it would be good to also include values (average ± SEM) that allow observation and comparison of the differences between groups
Response 7: We added the mean and SD for the groups that had statistical differences in the mother’s genotoxic analysis (lines 87-88, lines 92-93)
Comments 8: P2, lines 81, 85. In the text you refer to graphs in panel A and B, but in the referenced graphs the labels appear as lowercase a and b. This can cause some confusion, so I recommend homogenizing this nomenclature.
Response 8: It was changed in the text as lowercase a and b (Lines 93,109,123)
Comments 9: P3, fig a, b. I suggest that in both graphs, instead of organizing the X-axis with the 5 treatments (5 groups, each including the 6 time points), you organize it by the 6 sampling times (0, 24, 48, 72, 96, 120 h). Then, each bar within each time point would correspond to the 5 treatments (H2O, CP, and the 3 doses of Hs). This would help to more efficiently visualize the differences in frequencies between the groups at each time. Also, please indicate the dose of CP (CP 30 mg/kg), not just CP
Response 9: The comparisons are within-group, comparing baseline time (0h) with 24, 48, 72, 96, and 12h. This is why the results are represented this way, allowing for comparisons between baseline and other times in the same group. If we make the suggested changes, it would be visualized as if the comparisons were between groups. (figure 1)
Comments 10: P3, fig a, b. On the other hand, it is striking that in your experiment the frequency of PCEs and MNPCEs with a dose of 30 mg/kg of CP increases up to 96 h. While the literature reports that with 20 mg/kg of CP, a significant increase in both biomarkers can be observed as early as 48 h. Please develop this in the discussion section.
Response 10: Thanks for that comment, we administered CP only the gestational days 19 and 20 (72 and 96 h), and the increase in the biomarkers coincides with the administration time. (Lines 88 and 91)
Comments 11: P3, line 93. In the figure caption you indicate statistical differences using ** and *** associated with p-values. However, it is not clear what groups are being compared in the graph. It is more important to state that treatment 1 differed from treatment 2 than to say ** means p < 0.01 or *** means p < 0.001. That is, in the figure legend, you should just indicate p < 0.05 but describe more clearly which comparisons are represented by the asterisks or symbols. For example: statistical differences vs **H2O. This means that all bars with ** are statistically different from the H2O group.
Response 11: In the figure caption it was specified that Intragroup comparisons were performed between baseline samples (0 h) against the following sampling times: 24, 48, 72, 96, and 120 h. ** p ≤0.01, *** p ≤0.001 (lines 97-100)
Comments 12: P4, figure 2a, b, c. Please homogenize the nomenclature in the text and figures (either uppercase A, B, C or lowercase a, b, c in both). Also, do not just write CP, but CP 30 mg/kg in the graphs. And in the legend, describe in more detail the statistical differences between treated groups: define all statistical symbols and comparisons.
Response 12: The nomenclature in the text of the figures was homogenized (capital letters were changed to lower case), CP was changed to CP (30 mg/kg) in the graphs. In the figure captions are specified the comparisons made and the statistical analysis and symbols used. (Figure 2)
Comments 13: P4, lines 103-124. The description of your results shown in figure 2 would be clearer if you included numerical values and not just a description of trends when comparing the effect of the different treatments.
Response 13: We added the medians for every comparison in the mother’s MDA quantification (Lines 112,115,118)
Comments 14: P4, lines 127–129. This section lacks a lot of information for complete somatometric analysis. Starting with maternal–fetal parameters such as: number of mated females, number of pregnant females, pregnancy index, fertility index, maternal weight gain, females with live or dead newborns, live newborns, live newborns per litter, dead newborns per litter, etc. This would show the effects that the treatments had on gestation and the offspring.
Response 14: The pregnancy rate was 100% since all rats were pregnant, all neonates were alive, and the number of neonates per group (Lines 139-142) was added to the manuscript.
Comments 15: It is well documented that CP administration during rodent gestation produces serious structural malformations, growth retardation, and embryotoxicity. The absence of teratogenic effects in your results is the product of your experimental design (CP was administered during advanced developmental stages that did not allow teratogenic effects to occur). Therefore, you must justify in the methodology or discussion why you decided to omit this valuable information; as you said, there are no reports on the toxic effects of Hs, so the evaluation of its teratogenic effects (alongside the genotoxic ones) would have been of great value.
Response 15: We add in the methodology section the justification for the use of cyclophosphamide on days 19 and 20 of gestation (Lines 373-375), and we also mention the relevance of genetic damage in the occurrence of teratogenic effects in the introduction section (Lines 65-67).
Comments 16: P5, lines 147–149. In addition to explaining how the negative and positive controls (CP) behaved, it is also necessary to describe the behavior of the evaluated parameters in the three groups treated with Hs.
Response 16: We started describing the results from the controls and in the next paragraph we mentioned the results found in the groups administered with the Hs extract (Lines 164-174)
Comments 17: P5, table 1. The results of your table are interesting, but the way you indicate statistical differences is confusing. These differences should not only be mentioned in the table footnote, but also shown clearly in the table itself. I see you are using letters to denote differences, which is good. However, you are making it more complex than necessary. The description of your differences can be simpler. First, instead of placing the exact p-value in the table, just mention in the table footnote that p < 0.05. Second, place the letters where appropriate in each cell of the table. Then explain what those letters mean. For example, a clear and simple way would be: statistically significant differences (p < 0.05) vs aH2O, bCP 30 mg/kg, cHs 2000 mg/kg, d etc. In this statement, a means that anything with the letter a is statistically different from the H2O group, b from the CP 30 group, c from Hs 2000, and so on. This way of presenting the differences is much simpler and clearer. This would improve transparency and allow readers to assess statistical significance more precisely.
Response 17: We modified the table 1 to make the changes you suggest about how the p value is represented (Lines 152-156)
|
Comments 15: It is well documented that CP administration during rodent gestation produces serious structural malformations, growth retardation, and embryotoxicity. The absence of teratogenic effects in your results is the product of your experimental design (CP was administered during advanced developmental stages that did not allow teratogenic effects to occur). Therefore, you must justify in the methodology or discussion why you decided to omit this valuable information; as you said, there are no reports on the toxic effects of Hs, so the evaluation of its teratogenic effects (alongside the genotoxic ones) would have been of great value. Response 15: Thanks for the comment, I added in the methodology section the justification of the cyclophosphamide use in those specific days (Lines 373-375), also, we mentioned the relevance of the genetic damage in the appearance of teratogenic effects on the introduction section (Lines 65-66)
Comments 16: P5, lines 147–149. In addition to explaining how the negative and positive controls (CP) behaved, it is also necessary to describe the behavior of the evaluated parameters in the three groups treated with Hs. Response 16: Thanks for signaling that, we started describing the results from the controls and in the next paragraph we mentioned the results found in the groups administered with the Hs extract (Lines 164-169)
Comments 17: P5, table 1. The results of your table are interesting, but the way you indicate statistical differences is confusing. These differences should not only be mentioned in the table footnote, but also shown clearly in the table itself. I see you are using letters to denote differences, which is good. However, you are making it more complex than necessary. The description of your differences can be simpler. First, instead of placing the exact p-value in the table, just mention in the table footnote that p < 0.05. Second, place the letters where appropriate in each cell of the table. Then explain what those letters mean. For example, a clear and simple way would be: statistically significant differences (p < 0.05) vs aH2O, bCP 30 mg/kg, cHs 2000 mg/kg, d etc. In this statement, a means that anything with the letter a is statistically different from the H2O group, b from the CP 30 group, c from Hs 2000, and so on. This way of presenting the differences is much simpler and clearer. This would improve transparency and allow readers to assess statistical significance more precisely. Response 17: We modified the table 1 to add the p value for each comparisson as you suggested (Table 1) Comments 18: P5, lines 145–150. Before explaining your results, describe how your groups were formed: how many neonates were obtained in each litter, and therefore in each group. And explain why only 30 neonates per group were used (how were they selected?). Response 18: Thanks for the observation. The average number of newborns in each group was specified in Lines 143-145, and I added the explanation of how we obtained the sample size in Lines 144-145.
Comments 19: Also, you need to explain why in the neonates another parameter (MNEs) was determined that was not evaluated in the pregnant females. Response 19: Thanks for that observation, the number of MNEs was not counted in mothers, this is due to the spleen in adult mammals is fully functional and can easily delete the damaged erythrocytes [34,46]. (Lines 439-440)
Comments 20: P5, lines 157–161. You report genotoxic effects in neonates at all doses, but these were not observed in the mothers. So, this discrepancy warrants deeper discussion. For example, could developmental stage or pharmacokinetics explain the higher susceptibility? What do you think? Please explain this in the discussion section. Response 20: This behavior could be due to the fact that newborns have still immature xenobiotic elimination systems, and the activity of liver enzymes begins until days after birth, this delays the clearance of foreign compounds. (line 236-238)
Comments 21: P6, fig 3a. I find it striking that they present the data on genotoxicity in newborns in table format. When the same genotoxicity data in mothers was shown as bar graphs. I consider it very important that the structure of your results be consistent. Therefore, I believe it is necessary that the genotoxicity data for newborns be shown in the same format as the genotoxicity data for pregnant females. Not to mention that it is not necessary to show the P values in your table (or graph). As I have been explaining, just indicate the differences with letters or symbols and describe them in the footnote of the table/graph next to the P value <0.05. Response 21: We decided to present the results the way we did because the results from genotoxicity in mothers were parametric and the results from genotoxicity in newborns were not parametric, also, we modified the table a in figure 3 to present the p value as you suggested. (figure 3)
Comments 22: P6, fig3b, c, and d. Why are you using box and whisker plots in graphs b, c, and d, instead of bar graphs as you did with these same parameters in mothers? If you want to show the dispersion of the data, it would also be useful to do the same with the data for mothers in Figure 2. To do this, you can use bar charts that, in addition to the mean and standard error, also show the dispersion of the data in the form of dots. Comments 22: The data obtained had a non-parametric distribution, so we represented them in box and whiskers, we changed the graphs to present them as dispersion dots as you suggested (Figure 3) Comments 23: The data shown in graphs 3b, c, and d are the same as those in table 3a. It makes no sense to repeat the same data again in graph format. Especially since table a already shows the minimum and maximum values for each group in parentheses. Ergo, there is no new information that justifies the use of scatter plots that repeat the same data. Please standardize these results: remove Table 3a, and instead of box and whisker plots, use bar graphs, as you did with the genotoxicity data for pregnant females. Response 23: We deleted the table a in figure 3. Comments 24: P7, fig 4. Please address the same comments I made in figure 2. Response 24: We modified the letters that represent the panels as you suggested (Figure 4) Comments 25: P8, lines 212-218. The information in this section is background information that should be explained in more detail in the background section (to understand, based on the bioactivation of CP, how this molecule causes the formation of micronuclei and alterations in embryo-fetal development). In this section, rather than explaining the mechanisms of CP, they should only be briefly revisited to highlight or contextualize some of your findings. Remember that the discussion section focuses on comparing your results with those of other researchers, in order to explain, based on the scientific literature, the reasons for the differences or similarities found. Response 25: We deleted the information about the CP bioactivation from the discussion section, to the experimental design section to justify the use of CP (Lines 375-380)
Comments 26: P8, line 223. You mention that the Hs extract caused myelosuppression, as evidenced by a decrease in PCEs, but you do not mention at what point this occurred, which is extremely important information. Please add this information. Response 26: The dose of 2,000 mg/Kg did cause cytotoxicity at the 120 h, evidenced by a reduction in the number of PCEs. This decrease is associated with myelosuppression of the bone marrow. (line 226-227) Comments 27: In this context, it is necessary to mention that the statistical differences in the graph are very confusing; the asterisks on the 120-hour bar for Hs 2000 mg/kg do not indicate from whom they are different. And since the averages at each sampling time are similar to those at 120, the question is: compared to whom is the frequency of PCEs at 120 different in the Hs 2000 mg/kg group? I am not sure if there is actually myelosuppression, since the frequency of PCEs at 120 h is very similar to the frequency of PCEs at 0 h (physiological levels). Please explain your findings in detail. Response 27: thanks for that comment, in the legend of the figure we mentioned that all the comparisons were made vs the baseline sample (0h). The count of PCEs in the EHHs 2,000 mg/Kg at the 120h showed a decrement in 10.062% vs 0h and CP showed a decrement in 17.83% vs 0h, being both decrements statistically significant according to the repeated measures ANOVA test with a Bonferroni post-hoc. (figure 1) Comments 28: In addition, I believe it is necessary to discuss at length the fact that in your group treated with CP 30 mg/kg, no decreases or increases in the frequencies of PCEs and MNPCEs (respectively) were observed before 96 hours. When the scientific literature reports that at doses of 20 mg/kg, the decrease and increase in the frequency of these markers can be observed after 24 hours, reaching their peak between 36 and 48 hours post-treatment. It should also be noted that several articles report a marked normalization of these markers, leading to the frequency of these biomarkers being practically normal (physiological or basal) between 96 and 120 hours post-treatment. Response 28: Thank you for pointing that, we administered CP only the last 2 gestational days (72 and 96 h), that is why we observed the differences in the number of PCEs in the 96 hours. (figure 1) Comment 29: P8, line 224-128. It is very interesting that no genotoxic effects derived from Hs treatment were observed in pregnant females, while these effects were observed in neonates. Please expand your discussion by explaining what might be causing this phenomenon. Response 29: This behavior could be due to the fact that newborns have still immature xenobiotic elimination systems, and the activity of liver enzymes begins until days after birth, this delays the clearance of foreign compounds (line 236-238)
Comment 30: P8, line 229. The long-term genotoxicity observed only at the lowest dose of Hs (500 mg/kg) raises questions. Please discuss the possibility of a hormetic effect or non-monotonic dose-response. Response 30: Thank you for that observation, we added some information about the possible hormetic effect in the discussion section (Lines 233-235) Comment 31: P8, lines 235-245. You mention the molecules and background that may be responsible for the genotoxic and cytotoxic effects found after Hs administration. However, you need to explain the mechanism by which quercetin generates genotoxicity in a murine model and correlate it with your results. Response 31: In that study quercetin was isolated from Hs to evaluate its genotoxic effect with an AMES assay, the author found that quercetin gave positive genotoxic results, however they only recommend extending these findings to in vivo tests, without really addressing the mechanisms of genotoxicity, that is why we did not delve much into that study. (247-249) Comment 32: On the other hand, their results show that Hs was myelostimulant, but the studies they cite show that Hs is cytotoxic in other cells. This contradiction raises questions that need to be discussed and clarified. Response 32: The fact that the EHHs had a myelostimulating effect despite the fact that there is evidence of its cytotoxic effects, reinforces the existing information about the nature of hematopoietic cells, which are considered to have special mechanism of self-renewal and differentiation. (Lines 239-242) Comment 33: Finally, they also mention that Hs has reports of genoprotection against CP. This is interesting because it raises questions such as: if in your experiment Hs induced genotoxicity in neonates in the short and long term, how does this finding relate to previous reports that indicate that Hs has protective effects against CP-induced genotoxicity? Please expand your discussion on this important point. Response 33: Thank you for that suggestion, we added some information in the discussion section about how our results complement the previous studies. (Lines 252-257) Comment 34: P9, lines 260-270. The explanation of MDA levels and flavonoid saturation is intriguing but needs stronger support. Please consider citing experimental studies with similar conditions to support this mechanistic hypothesis. Response 34: Thank you for that comment, we added only in vitro studies because we did not find in the literature consulted any in vivo or clinical study about the prooxidant effects of flavonoids (290-295) Comment 35: P9, lines 273-282. Consider adding a section to your discussion that delves into the experimental strengths and limitations, contextualizes your findings, and directs those interested in continuing this research to the most important aspects that should be explored. Response 35: We added a final section in the discussion talking about the strengths and limitations of our study (Lines 316-330) Comment 36: P9, lines 283–303. Your extract preparation protocol is thorough. However, you need to provide more technical details so that anyone interested can repeat your methodology: at what refrigeration temperature was the macerate stored? What were the conditions of temperature, reduced pressure, and time under which the extract was concentrated in the rotary evaporator? Why did you concentrate the extract in a rotary evaporator before freeze-drying, rather than freeze-drying it directly? Is this important in the context of metabolite stability? You should mention whether the chemical stability of the extract was evaluated over time, as this could influence metabolite bioavailability. What were the freeze-drying conditions used? In addition, please cite recent studies that support each procedure used in your methodology. It is also necessary to explain the procedures used to carry out the phytochemical analysis. Consider adding the results of this analysis to the results section, as this will help to better contextualize your findings. Response 36: We added more information about the EHHs preparation as you suggested, we concentrated the extract in the rotary evaporator to evaporate the ethanol from the mixture and just get the water, then we freeze-dry just the water part, also, we decided to add phytochemical analysis in the supplementary material because we decided not to focus our study on phytochemical issues but focus in the genotoxic and cytotoxic evaluation. (235-255) Comments 37: P10, lines 305-335. Specify in your design that the n for each group of females was 5 (n=5). Although OECD 474 (Micronucleus Test) indicates that n=5 is sufficient to demonstrate genotoxic effects, consider that similar studies typically use at least n=6 to ensure the statistical robustness of the results. And that OECD-414 recommends n=15 for teratogenicity tests. Therefore, it is important to justify the use of n=5 in your design; cite research that supports your design. Comments 38: Please discuss the possible limitation of having n=5 per group, and only six neonates per mother per group. How might this sample size affect the generalizability of your findings? Response 37 y 38: The sample size in this study was determined based on the guidelines established in the protocol for the micronucleus assay in mammals, which specifies that each experimental group should consist of a minimum of 5 animals. Although previous genotoxicity studies commonly include six or more animals per group, the present study deliberately employed the minimum required number in order to adhere to the Reduction principle of the 3Rs (Replacement, Reduction, and Refinement). This decision aligns with the recommendations of the OECD, which advocate for minimizing the use of animals in scientific research without compromising the validity and reliability of the results. [34,60]. (Lines 364-371) Comments 39: P10, table 2. Why did you use Tridistilled H2O as a negative control? Although it is not toxic, it is known that the lack of ions in ultrapure water can alter the cellular osmotic balance. While it is true that the OECD does not prohibit its use, some agencies (EMA, FDA) recommend standard vehicles such as purified water or saline. Please explain. Response 39: The use of triple-distilled water in our study was based on its high purity, in addition to the fact that a low volume is administered, only 0.1 ml per 10 g of weight per day. Furthermore, the animals consume purified water ad libitum from their drinking troughs, thereby recovering the lack of ions. (Lines 371-373) Comments 40: P10 lines 322-326. The presence of sperm only indicates that mating took place, but does not guarantee that the mated females are pregnant. To reduce the margin of error, in addition to the presence of sperm, it is necessary to demonstrate that copulation took place during the fertile phase of the rat's estrous cycle. Therefore, the identification of early estrus, estrus, or late estrus during mating is necessary. Please supplement your information. Response 40: We matted the rats during the estrus phase, which we confirmed with a cytological analysis (lines 398-399) Comments 41: P10, 327-328. In this section, you should explain why the compounds were administered at a late stage of development, when alterations are unlikely to occur, rather than during sensitive periods of development, in order to guarantee the absence of teratogenic effects of the extract, as suggested by the OECD. Although your research focuses on genotoxic effects, given your experimental design that includes intrauterine development, it is necessary to clarify these methodological details to ensure the correct use of experimental animals for maximum benefit. Response 42: The different compounds were administered only during the last 5 days of gestation because, after exposure to EHHs, any chromosomal damage in erythrocytes becomes visible in the PCEs that appear in peripheral blood approximately 24–48 h after the injury and stabilizes in circulating erythrocytes at 72 h. Sampling during the last 5 days ensures that sufficient time has elapsed between the first day of administration and the final sample for micronuclei to form and be detected [34,36,61]. (Lines 388-393) Comments 43: P10, lines 333-336. Explain why only six neonates per litter were selected and what the selection criteria were. Clearly describe the condition of the neonates obtained for the corresponding determinations and when the blood samples were taken. Cite research that supports your design. Response 43: Thank you for the advice, each treated and control group should include at least five animals per group, and at least four analyzable animals in accordance with the guidelines established for performing the MN test in rodent blood. The teratogenesis model used in this study is based on a previously established model described by Gómez-Meda et al., (doi:10.1002/em.20037.) in which the unit of analysis in this study is the pregnant rat; therefore, the group size is determined based on the number of these animals. (Gomez-Meda 2004) The effects of the extracts administered to the mothers during gestation will be evaluated in the neonates. The newborns were selected per litter by simple randomization; the paragraph was modified to avoid confusion (410-413) Comments 44: P10, lines 338-345. The use of ether as an anesthetic is no longer considered ethical or recommended due to its respiratory irritability, risk of explosion, and prolonged suffering in animals. Exsanguination euthanasia procedures are recommended when cleaning, fixing, and tissue removal for histological analysis purposes are justified. If this is not to be done, there are much simpler procedures in adult rats for ethical euthanasia, such as cervical dislocation, ketamine + xylazine, sodium pentobarbital, etc. For fetuses and neonates, ether as an anesthetic or for euthanasia purposes is also unethical. Instead, pentobarbital, controlled hypothermia, or decapitation (if justified by the methodology) are recommended. To say that a newborn has died, the absence of stimuli is not enough, as it may only be anesthetized. Please describe in detail how you ensured the death of the newborns. Response 44: Our sacrificing protocol was elaborated from the current regulation for animal use in Mexico and also with the support an approval of the Research Ethics Committee from the Autonomous University of Zacatecas’s nursing school, we needed to obtain as much blood as possible, which meant that the animals' anesthesia had to be kept under control. For this reason, ether was used as an anesthetic, as it anesthetizes the animal in seconds and the degree of anesthesia can be controlled with an ether chamber. The death of newborns was established with cessation of respiratory movements and an evaluation of the response to neurological reflexes ; this information was added to the main text. (Lines 417-420). Comments 45: Your methodology mentions that the pregnant females were anesthetized with ether. This is confusing, since your methodology mentions that the pregnancy was carried to term, which means that all the offspring were born and no female should have been pregnant at the time of sacrifice. Please correct or explain what you mean by pregnant females. Response 45: Thank you for that observation, it was a mistake, we corrected the word in the main text (Line 413) Comments 46: P11, 368-362. Why were the same parameters not evaluated in the genotoxicity study of mothers and newborns? I understand that micronucleated erythrocytes (MNE) refer to mature erythrocytes in which long-term genotoxicity was determined in newborns. Therefore, it is necessary to describe them in this way, referring to them as micronucleated normochromic erythrocytes. The term MNE is very general (it is not known whether they are young or mature erythrocytes) and can be confusing for people who are not in the field. Explain why MNE in neonates was determined in 10,000 cells and not in 1,000 as in PCEs. This information is important for understanding your methodology. Response 46: Thank you for that comment, in adult rats, the MNEs should not be evaluated because their spleen can easily remove them and would not give us reliable information about genotoxicity, also, the frequency of MNEs in TEs is low; to detect statistically significant increases compared to the control, it is necessary to increase the number of cells evaluated. Counting 10,000 total erythrocytes allows to reduce the sampling error and increase the statistical power of the assay. In addition, PCEs represent young cells in which genotoxic damage manifests most clearly shortly after the injury. Their proportion in peripheral blood is relatively low, so counting 1,000 PCEs is sufficient to obtain a reliable estimate of the frequency of micronuclei in this subpopulation. (Lines 440-441) Comment 47: P11, lines 380-382. The description of your statistical analysis is very limited; simply stating that X analysis was used for parametric variables and Y analysis for non-parametric variables is very limited. Instead, please provide more information and specify the data or variables analyzed with certain tests. For example, in the micronucleus assay, it is not advisable to use repeated measures analysis of variance (ANOVA) that compares only with the baseline (0 h), as it ignores interactions between time and treatment. It is more robust to use a two-way ANOVA for repeated measures, as they have two factors: treatments and sampling days. Response 47: The Shapiro-Wilk and Kolmogorov–Smirnov test were used to establish that the data were normally distributed. The ANOVA repeated measures statistical analysis was used to compare the means of the dependent variables EPC and EPCMN at different sampling times in the same group, since we are interested in knowing if there are increases or decreases in these markers with respect to the baseline value. For newborn rats, the litter was used as the experimental unit (n = 6/group), and pups were the unit of observation and analysis. Intergroup comparisons were made using the Kruskall–Wallis analysis with Dunn’s post hoc. Data for MDA concentrations were expressed as a median with the maximum and minimum. Intergroup comparisons were made using the Kruskall–Wallis analysis with Dunn’s post hoc (457-472) Comment 48: P12, lines 388-393. I recommend softening the conclusion. Instead of discouraging use entirely, consider suggesting that further evaluations are necessary and that caution should be exercised during pregnancy. Also mention the methodological or approach limitations of your study. Response 49: Thank you for the advice, we reformulate the conclusion section to include these recommendations. (Lines 474-481).
Minor comments 1. Review and maintain consistency in acronyms such as: EHHs vs EEHs, PCEs vs PCE, etc. 2. Carefully review the use of abbreviations, as there are terms that you show in parentheses at the beginning of your manuscript and then show again in parentheses in the last sections of your document. 3. Standardize the way you present figures; for example, sometimes you write 2,000 and other times 2000. 4. Try to reduce passive voice to improve readability. 5. Ensure all references are correctly formatted and up to date. Response: Thank you so much for these recommendations, we checked the manuscript to apply all these changes. |
|
|

Reviewer 2 Report
Comments and Suggestions for Authors
The study's findings are valuable; however, several issues in each section need to be significantly addressed.
Abstract:
- Page 1, line 17; The phrase "it is mentioned that (ere assessed in Wistar rats and their n)" is unclear and seems incomplete or incorrectly structured. Clarify what n refers to.
- Add the route of administration and the number of rats in each group in the abstract.
Introduction:
Page 2, lines 46 and 47: mention studied group animal or human (similarly, doses of 200 mg/Kg of aqueous extracts obtained via cold and hot extraction caused sperm morphological abnormalities in 43.5% 47 and 52.5% of cases, respectively):
Line 55: The current study did not assess teratogenic impact. Please correct (For this reason, the present study focuses on assessing the potential genotoxic, cytotoxic, and teratogenic effects of the hydroalcoholic extract of Hs)
Materials and Methods:
- The source of plant material from Sam’s Club® raises questions about plant verification. How are you sure that you chose the right plant?
- Although analysis was mentioned in the supplementary section, it is better to add a summary in the method with add in-text citation of supplementary files to guide readers.
- In the extraction protocol, please mention the reference to this protocol.
- How do you detect supernatant concentration, as it is the cornerstone of your experiment? It mentioned that (Every 24 hours, three times, the macerate was filtered. The supernatant was stored under refrigeration and shielded from light, while the residual plant material was returned to the flask, and fresh solvent was added.
- "Medium porosity filter paper" is vague. Specify pore size :(After completing the maceration cycles, all of the collected supernatant was combined and filtered through medium porosity filter paper.)
- What is the target PH level in your study, as mentioned that (the extract’s pH was measured using a pH)
- Dosing Rationale: Why 2,000, 1,000, 500 mg/kg? Are these based on prior toxicity studies or traditional use? Please add the dose rationale with references.
- MDA reflects lipid peroxidation but not direct cell death (Indirect Cytotoxicity Marker). MDA is useful but insufficient alone for cytotoxicity evaluation. Why did you not use other cytotoxic parameters?
- Please add clarification of the Lee index in the result section with interpretation
- Discuss why a large dose of 2000 mg/kg induces less MDA level than a 1000 mg/kg dose.
Author Response
Response to Reviewer 2 Comments
Thank you very much for taking the time to review this manuscript. Below, you will find detailed responses to comments made to the submitted manuscript, changes made have been highlighted in red and the file has been resubmitted.
Point-by-point response to Comments and Suggestions for Authors
Comments 1: Page 1, line 17; The phrase "it is mentioned that (ere assessed in Wistar rats and their n)" is unclear and seems incomplete or incorrectly structured. Clarify what n refers to. Add the route of administration and the number of rats in each group in the abstract.
Response 1: Thank you for pointing that, it was a mistake while typing, which we already corrected (lines 18)
Comment 2: Page 2, lines 46 and 47: mention studied group animal or human (similarly, doses of 200 mg/Kg of aqueous extracts obtained via cold and hot extraction caused sperm morphological abnormalities in 43.5% 47 and 52.5% of cases, respectively):
Response 2: It was added that in the group in which sperm morphological abnormalities occur at a dose of 200 mg/kg the aqueous extracts obtained by hot and cold extraction are in “rats” (Lines 49-50).
Comments 3: Line 55: The current study did not assess teratogenic impact. Please correct (For this reason, the present study focuses on assessing the potential genotoxic, cytotoxic, and teratogenic effects of the hydroalcoholic extract of Hs)
Response 3: The word teratogenic was deleted (line 73)
Comments 4: The source of plant material from Sam’s Club® raises questions about plant verification. How are you sure that you chose the right plant?
Response 4: Hibiscus sabdariffa calyxes were obtained in a single lot from SAM's as a prepackaged food that complies with the Mexican official standard NOM-051-SCFI/SSA1-2010 General labeling specifications for prepackaged food and non-alcoholic beverages -Commercial and sanitary information (336-337)
Comments 5: Although analysis was mentioned in the supplementary section, it is better to add a summary in the method with add in-text citation of supplementary files to guide readers.
Response 5: Quotes have been added to the text, as follows:
Table S1: Identified compounds in the EHHs
- Total anthocyanin content
- Total phenolic content
- Total flavonoid content
- Identified of secondary metabolites using a gas chromatography–mass spec-trometry system (line 507-512)
Comments 6: In the extraction protocol, please mention the reference to this protocol.
Response 6: The reference was added (335-336)
Comments 7: How do you detect supernatant concentration, as it is the cornerstone of your experiment? It mentioned that (Every 24 hours, three times, the macerate was filtered. The supernatant was stored under refrigeration and shielded from light, while the residual plant material was returned to the flask, and fresh solvent was added.
Response 7: Three extractions of the plant material are made, the plant material is placed under mechanical maceration for 24 hours and filtered, the filtrate is recovered and stored refrigerated protected from light, again the filtered plant material is macerated again with fresh solvent in the same proportion for 24 hours, this is done two more times. The supernatant or solvent resulting from the maceration that was stored in refrigerated is concentrated in a rotary evaporator to eliminate methane and only the aqueous part remains, subsequently it is frozen and lyophilized, leaving a powder which can be weighed.
Comments 8: "Medium porosity filter paper" is vague. Specify pore size:(After completing the maceration cycles, all of the collected supernatant was combined and filtered through medium porosity filter paper.)
Comments 8: We added that the pore size is 8μm (line 349)
Comments 9: What is the target PH level in your study, as mentioned that (the extract’s pH was measured using a pH)
Response 9: We added that the target pH was 2.2. (line 350)
Comments 10: Dosing Rationale: Why 2,000, 1,000, 500 mg/kg? Are these based on prior toxicity studies or traditional use? Please add the dose rationale with references.
Response 10: The doses used in our study were chosen in accordance with the OECD mammalian micronucleus protocol, so the following paragraph was added: The doses of the EHHs administered in this study were determined based on the protocol for the micronucleus assay in mammals, which states that, in the absence of prior genotoxicity data for the test substance, the highest dose used should be 2,000 mg/Kg. Subsequent doses should be established by reducing the previous one by 50%, in a stepwise manner. [34]. (Lines 381-385)
Comments 11: MDA reflects lipid peroxidation but not direct cell death (Indirect Cytotoxicity Marker). MDA is useful but insufficient alone for cytotoxicity evaluation. Why did you not use other cytotoxic parameters?
Response 11: Thank you for that comment, the choice of MDA as a marker of cellular damage was based on its usefulness as an early and sensitive indicator of oxidative stress, a process that has been reported as one of the main mechanisms involved in induced toxicity (https://doi.org/10.1155/2014/360438), in addition, it is also determined whether the number of polychromatic erythrocytes decreased as a marker of myelosuppression, which is also a parameter of cytotoxicity.
Comments 12: Please add clarification of the Lee index with interpretation
Response 12: Thank you for that suggestion, the interpretation of the Lee index was added in the discussion section (Lines 269-271)
Comments 13: Discuss why a large dose of 2000 mg/kg induces less MDA level than a 1000 mg/kg dose.
Response 13: Although well-established antioxidant mechanisms of EHHs have been recognized, there is also in vitro and in vivo evidence that molecules such as flavonoids can exhibit pro-oxidant behavior, thus increasing oxidative stress under certain conditions, such as alkaline pH and the presence of transition metals such as Fe and Cu, among others. This could be the case for the observed behavior, which at a dose of 1000 mg/kg shows a prooxidant effect (Lines 290-295)

Reviewer 3 Report
Comments and Suggestions for Authors
This manuscript explores the effects of a hydroalcoholic extract of Hibiscus sabdariffa administered during late gestation in a rodent model. While the study aims to address an important question concerning maternal and neonatal safety of traditionally used herbal preparations, there are several significant methodological, analytical, and interpretive issues that limit the rigor and translational relevance of the findings.
The study employs doses up to 2,000 mg/kg without clear justification or reference to traditional or therapeutic use levels in humans. There is no attempt to contextualize these doses with respect to commonly consumed forms, such as tea or dietary supplements.
Maternal exposure was restricted to gestational days 16–20. This late gestational window excludes earlier developmental stages critical for organogenesis and genomic imprinting. The omission of early pregnancy exposure significantly limits conclusions about developmental toxicity. Moreover, assessments are confined to neonatal outcomes at birth, with no follow-up on postnatal development, behavior, or reproductive capacity—key components for evaluating long-term toxicological effects.
The rationale behind specific timepoints chosen for genotoxicity assessment (e.g., 24–120 hours) and the timing of neonatal evaluation is unclear and should be elaborated.
The neonatal sample size is notably small (only six pups in total), which severely restricts statistical power and generalizability. Additionally, there is no clarity on how these neonates were selected (e.g., per litter or across litters), nor any evidence that litter effects were accounted for in the statistical analysis. Given the known issue of non-independence in offspring data from the same dam, failure to correct for litter effects may lead to pseudo-replication and inflate the significance of findings. Furthermore, substantial variability in litter size between groups (e.g., 7.6 in treated vs. 12.2 in controls) could confound outcome interpretation.
The extract used was hydroalcoholic, but no control group received ethanol alone. This omission impedes the ability to distinguish effects attributable to the extract from those due to the solvent vehicle. A properly matched vehicle control is essential for accurate interpretation of results in extract-based studies.
The manuscript reports that non-parametric data were analyzed using repeated-measures ANOVA, which is inappropriate. ANOVA assumes parametric data and independence; suitable non-parametric alternatives (e.g., Friedman test or Kruskal-Wallis) should be applied. There is also no mention of how data assumptions—normality or homogeneity of variance—were tested. Some results (e.g., MDA levels) show no consistent dose-response relationship and include trends that are not statistically significant; however, these are occasionally overstated in the discussion.
The MDA (malondialdehyde) assay employed is based on thiobarbituric acid reactivity, which lacks specificity and is known to detect a range of aldehydes and oxidized lipids. This methodological limitation is not acknowledged. Moreover, instrument parameters for MDA quantification (e.g., wavelength, slit width, reading mode) are not sufficiently described to allow replication.
The plant material was sourced from a commercial retail outlet (Sam’s Club®), which does not ensure botanical authentication or consistency in chemotype. The lack of a voucher specimen and absence of phytochemical characterization undermine reproducibility. Additionally, the extract’s yield (w/w) from the dried plant material is not reported, which further limits standardization and comparability across studies.
While the study reports both beneficial (e.g., stimulation of erythropoiesis) and adverse (e.g., increased micronucleated polychromatic erythrocytes, MNPCEs; elevated MDA) effects at the same extract doses, the discussion does not attempt to reconcile this apparent duality. Nor does it explore potential mechanisms such as oxidative stress or specific bioactive compounds in H. sabdariffa that could underlie the observed effects. Furthermore, the toxicological significance of elevated MDA levels is not discussed in terms of tissue damage or long-term consequences.
Despite being a preclinical study with implications for maternal herbal medicine use, the manuscript does not extrapolate the findings to human contexts. There is no discussion on dose equivalence, potential safety thresholds, or the risk-benefit balance of H. sabdariffa consumption during pregnancy. This omission limits the translational impact of the research.
Several grammatical and typographical errors (e.g., “their n” instead of “their newborns”) detract from the manuscript’s clarity.
Author Response
Response to Reviewer 3 Comments
Thank you very much for taking the time to review this manuscript. Below, you will find detailed responses to comments made to the submitted manuscript, changes made have been highlighted in red and the file has been resubmitted.
Point-by-point response to Comments and Suggestions for Authors
Comments 1: The study employs doses up to 2,000 mg/kg without clear justification or reference to traditional or therapeutic use levels in humans. There is no attempt to contextualize these doses with respect to commonly consumed forms, such as tea or dietary supplements.
Response 1: The doses used in our study were chosen in accordance with the OECD mammalian micronucleus protocol, so the following paragraph was added: The doses of the EHHs administered in this study were determined based on the protocol for the micronucleus assay in mammals, which states that, in the absence of prior genotoxicity data for the test substance, the highest dose used should be 2,000 mg/Kg. Subsequent doses should be established by reducing the previous one by 50%, in a stepwise manner [34] (Lines 381-385)
Comments 2: Maternal exposure was restricted to gestational days 16–20. This late gestational window excludes earlier developmental stages critical for organogenesis and genomic imprinting. The omission of early pregnancy exposure significantly limits conclusions about developmental toxicity. Moreover, assessments are confined to neonatal outcomes at birth, with no follow-up on postnatal development, behavior, or reproductive capacity—key components for evaluating long-term toxicological effects.
Response 2: Maternal exposure was restricted to gestational days 16–20. This late gestational window excludes earlier developmental stages critical for organogenesis and genomic imprinting. The omission of early pregnancy exposure significantly limits conclusions about developmental toxicity. Moreover, assessments are confined to neonatal outcomes at birth, with no follow-up on postnatal development, behavior, or reproductive capacity—key components for evaluating long-term toxicological effects (Lines 388-393)
Comments 3: The rationale behind specific timepoints chosen for genotoxicity assessment (e.g., 24–120 hours) and the timing of neonatal evaluation is unclear and should be elaborated.
Response 3: The following explanation was added: The different compounds were administered only during the last 5 days of gestation because, after exposure to EHHs, any chromosomal damage in erythrocytes becomes visible in the PCEs that appear in peripheral blood approximately 24–48 h after the injury and stabilizes in circulating erythrocytes at 72 h. Sampling during the last 5 days ensures that sufficient time has elapsed between the first day of administration and the final sample for micronuclei to form and be detected [34,36,61]. (line 388-393)
Comments 4: The neonatal sample size is notably small (only six pups in total), which severely restricts statistical power and generalizability. Additionally, there is no clarity on how these neonates were selected (e.g., per litter or across litters), nor any evidence that litter effects were accounted for in the statistical analysis. Given the known issue of non-independence in offspring data from the same dam, failure to correct for litter effects may lead to pseudo-replication and inflate the significance of findings. Furthermore, substantial variability in litter size between groups (e.g., 7.6 in treated vs. 12.2 in controls) could confound outcome interpretation.
Response 4: Thank you for the advice, each treated and control group should include at least five animals per group, and at least four analyzable animals in accordance with the guidelines established for performing the MN test in rodent blood. The teratogenesis model used in this study is based on a previously established model described by Gómez-Meda et al., (doi:10.1002/em.20037.) in which the unit of analysis in this study is the pregnant rat; therefore, the group size is determined based on the number of these animals. (Gomez-Meda 2004) The effects of the extracts administered to the mothers during gestation will be evaluated in the neonates. The newborns were selected per litter by simple randomization; the paragraph was modified to avoid confusion (410-413).
Comments 5: The extract used was hydroalcoholic, but no control group received ethanol alone. This omission impedes the ability to distinguish effects attributable to the extract from those due to the solvent vehicle. A properly matched vehicle control is essential for accurate interpretation of results in extract-based studies.
Response 5: In the present study, the ethanol used during the extraction process was completely removed using a rotary evaporator under reduced pressure as we indicated in the methodology (lines 350-351). As a result, the administered preparation did not contain ethanol, allowing us to rule out any effects attributable to the solvent. Therefore, it was not considered necessary to include an ethanol control group.
Comments 6: The manuscript reports that non-parametric data were analyzed using repeated-measures ANOVA, which is inappropriate. ANOVA assumes parametric data and independence; suitable non-parametric alternatives (e.g., Friedman test or Kruskal-Wallis) should be applied. There is also no mention of how data assumptions—normality or homogeneity of variance—were tested. Some results (e.g., MDA levels) show no consistent dose-response relationship and include trends that are not statistically significant; however, these are occasionally overstated in the discussion.
Response 6: The Shapiro-Wilk and Kolmogorov–Smirnov test were used to establish that the data were normally distributed. The ANOVA repeated measures statistical analysis was used to compare the means of the dependent variables EPC and EPCMN at different sampling times in the same group, since we are interested in knowing if there are increases or decreases in these markers with respect to the baseline value. For newborn rats, the litter was used as the experimental unit (n = 6/group), and pups were the unit of observation and analysis. Intergroup comparisons were made using the Kruskall–Wallis analysis with Dunn’s post hoc. Data for MDA concentrations were expressed as a median with the maximum and minimum. Intergroup comparisons were made using the Kruskall–Wallis analysis with Dunn’s post hoc
Comments 7: The MDA (malondialdehyde) assay employed is based on thiobarbituric acid reactivity, which lacks specificity and is known to detect a range of aldehydes and oxidized lipids. This methodological limitation is not acknowledged. Moreover, instrument parameters for MDA quantification (e.g., wavelength, slit width, reading mode) are not sufficiently described to allow replication.
Response 7: The methodology establishes that the organic phase was read using a Jenway UV/Vis 6715 spectrophotometer (Cole-Parmer, IL, USA) at a wavelength of 534 nm. The level of MDA in the samples was determined using a calibration curve with 1,1,3,3-tetramethoxypropane (Sigma Chemical, St. Louis, MO, USA). (lines 443-455)
Comments 8: The plant material was sourced from a commercial retail outlet (Sam’s Club®), which does not ensure botanical authentication or consistency in chemotype. The lack of a voucher specimen and absence of phytochemical characterization undermine reproducibility. Additionally, the extract’s yield (w/w) from the dried plant material is not reported, which further limits standardization and comparability across studies.
Response 8: Hibiscus sabdariffa calyxes were obtained in a single lot from SAM's as a prepackaged food that complies with the Mexican official standard NOM-051-SCFI/SSA1-2010 General labeling specifications for prepackaged food and non-alcoholic beverages -Commercial and sanitary information both national and foreign manufacture, which are marketed in Mexico.
Comments 9: While the study reports both beneficial (e.g., stimulation of erythropoiesis) and adverse (e.g., increased micronucleated polychromatic erythrocytes, MNPCEs; elevated MDA) effects at the same extract doses, the discussion does not attempt to reconcile this apparent duality. Nor does it explore potential mechanisms such as oxidative stress or specific bioactive compounds in H. sabdariffa that could underlie the observed effects. Furthermore, the toxicological significance of elevated MDA levels is not discussed in terms of tissue damage or long-term consequences.
Response 9: The following paragraphs were added in discussion:
Although well-established antioxidant mechanisms of EHHs have been recognized, there is also in vitro and in vivo evidence that molecules such as flavonoids can exhibit prooxidant behavior, thus increasing oxidative stress under certain conditions, such as alkaline pH and the presence of transition metals such as Fe and Cu, among others. This could be the case for the observed behavior, which at a dose of 1000 mg/kg shows a prooxidant effect (line 290-294)
The absence of genotoxic effects in mothers, despite increased MDA levels, suggests that the oxidative stress induced by the EHHs was sufficient to alter lipid structures in metabolically active tissues such as the liver and kidneys, but did not reach a threshold capable of inducing chromosomal breaks or bone marrow dysfunction. It is therefore like-ly that the distribution and metabolism of EHHs led to a localized oxidative stress, with-out significant or sustained exposure of the DNA. (line 305-309)
Comments 10: Despite being a preclinical study with implications for maternal herbal medicine use, the manuscript does not extrapolate the findings to human contexts. There is no discussion on dose equivalence, potential safety thresholds, or the risk-benefit balance of H. sabdariffa consumption during pregnancy. This omission limits the translational impact of the research.
Response 10: We recognize the limitations of the project; however, the findings of this study provide valuable insight into the current knowledge gap regarding the safety profile of Hs during critical physiological stages such as gestation and fetal development, which could help guide future evaluations in human populations.
Comments 11: Several grammatical and typographical errors (e.g., “their n” instead of “their newborns”) detract from the manuscript’s clarity.
Response 11: The requested grammatical and typographical review is carried out.

Round 2
Reviewer 1 Report
Comments and Suggestions for Authors
Dear Authors,
Thank you for submitting the revised version of your manuscript. However, after carefully reviewing the updated document, I must express my concern regarding the lack of substantial improvement in response to the initial review.
In my first evaluation, I provided you with a comprehensive and detailed set of comments addressing key aspects of your study—methodology, statistical analysis, data presentation, and scientific discussion. I also offered specific guidance on how to address each of these points. Unfortunately, the revised manuscript does not reflect meaningful progress, and many of the original concerns remain unresolved.
There are still numerous issues in your methodology, results, and discussion that have not been adequately addressed:
- Several critical data points and statistical values I requested are still missing.
- Methodological decisions such as the use of ether, the selection of six neonates per dam, and the use of tridestilled water as a control remain poorly justified or unsupported by references.
- The low frequency of micronuclei in mothers is not explained, nor is the discrepancy between maternal and neonatal genotoxicity explored in depth.
- The discussion lacks the depth and scientific rigor necessary to interpret your findings in the context of existing literature.
- The manuscript has only undergone superficial edits, while the core scientific and methodological limitations persist.
- The relationship between genotoxicity and teratogenicity is completely omitted.
- The methods of euthanasia are unethical.
- And many more.
I invested significant time and effort in reviewing your first submission with the intention of helping you improve your work. Regrettably, I do not see that effort reflected in the revised version. The manuscript, in its current form, continues to exhibit the same fundamental weaknesses. I cannot offer any further comments or suggestions, as you have not adequately addressed those I made initially.
Best regards
Author Response
Response to reviewer 1 comments (2nd round)
We sincerely thank you for the time and effort you have dedicated to reviewing our manuscript. We greatly appreciate your insightful comments and constructive feedback. We will carefully follow your recommendations to improve the clarity, structure, and overall quality of the manuscript. Your observations have been invaluable in helping us enhance the scientific rigor and presentation of our work, and we remain committed to addressing each point thoroughly in our revised submission.
Comments 1: Several critical data points and statistical values I requested are still missing
Response 1: Among your suggestions was the following: “P3, fig a, b. I suggest that in both graphs, instead of organizing the X-axis with the 5 treatments (5 groups, each including the 6 time points), you organize it by the 6 sampling times (0, 24, 48, 72, 96, 120 h). Then, each bar within each time point would correspond to the 5 treatments (H2O, CP, and the 3 doses of Hs). This would help to more efficiently visualize the differences in frequencies between the groups at each time. Also, please indicate the dose of CP (CP 30 mg/kg), not just CP”. The image was not modified according to your suggestion because, as previously explained, its purpose is to illustrate the genotoxic and cytotoxic effects of a xenobiotic in this case, Hs using a negative control (water) and a positive control (CP) through the micronucleus assay. To determine the genotoxicity and cytotoxicity of a xenobiotic using this assay, it is necessary to compare the means of the dependent variable (either PCMNEs or PCEs) at different time points (24, 48, 72, 96, and 120 hours) relative to the baseline value. For this reason, a repeated measures ANOVA is applied, as each animal serves as its own control. Consequently, the treatment is represented across six time points, since this is an intragroup analysis. The approach you suggest would correspond to an intergroup analysis, which does not apply in this case, as only the 2,000 mg/kg dose showed a genotoxic effect. Therefore, there is no other Hs dose to compare it with in order to conduct an intergroup analysis.
Response 1: also, you mentioned in comment 13 that we should add numeric values on figure 2, so we added the P value of the different comparisons inside the figure 2.
Response 1: In comment 14 you suggest the addition of some somatometric parameters, so we added the next paragraph in the main text on lines 142-144 “The number of mated females was 25, we obtained a pregnancy rate of 100%, with a fertility index of 100%, similarly, the death of any of the newborns of the evaluated groups was not presented, obtaining total of 254 born alive newborns”. You also suggested that we add data about the weight gain of mothers, however, we do not have that data, therefor we did not add it to the main text.
Comments 2: Methodological decisions such as the use of ether, the selection of six neonates per dam, and the use of tridestilled water as a control remain poorly justified or unsupported by references.
- The selection of six newborns per litter (three males and three females) was carried out using simple randomization, following the methodology established by Gómez-Meda et al. (2004). Folate supplementation of cyclophosphamide-treated mothers diminishes micronucleated erythrocytes in peripheral blood of newborn rats. Environmental and molecular mutagenesis, 44(2), 174–178. https://doi.org/10.1002/em.20037. We added the next justification in lines 447-449.
- Triple-distilled water is used because it undergoes a distillation process three times, which eliminates most impurities such as salts, minerals, microorganisms, and organic compounds. This ensures that the water is as pure as possible and does not introduce additional variables into the study. It also does not affect the animal's osmotic balance, since the volume administered is only 0.1 mL per 10 g of the animal's weight once a day.
- We used ether as anesthetic because in Mexico, the animal sacrifice is governed by Mexican Official Standard NOM-033-SAG/ZOO-2014, Methods for killing domestic and wild animals. On section 11, entitled Prohibited Methods, establishes the following: “Any method or substance not authorized in this list is prohibited: The use of substances that induce muscle paralysis without causing loss of consciousness and that cause death by asphyxiation is prohibited. These substances are: succinylcholine, curare, strychnine, galamine, and pancuronium, as well as their derivatives. The use of succinylcholine and curare is only authorized in crocodiles and megavertebrates in emergency cases; The use of potassium chloride (KCl) in any form to cause the death of animals is also prohibited, as its administration causes intense pain and anxiety, followed by cardiac arrest in diastole in the conscious individual. Its use is only authorized for megavertebrates, provided that the animal is under deep anesthesia and this is verified by a veterinarian; The use of hypothermia and electricity is prohibited for stunning, anesthesia, slaughter, and euthanasia of all reptiles; It is prohibited to kill rodents, lagomorphs, and small mammals by hypothermia and/or freezing, by chest compression, by strangulation, by drowning, by air embolism, or by other mechanical methods of asphyxiation”. Since ether is not among the methods prohibited by Mexican regulations, and in accordance with the regulations governing the operation of the rodent vivarium at the Autonomous University of Zacatecas which says that the recommended inhalable anesthetics for euthanasia of animals weighing less than 7 kg, or in those in which venipuncture is difficult, are: halothane, enflurane, isoflurane, methoxyflurane, and ether, alone or in combination with nitrous oxide.
Comments 3: The low frequency of micronuclei in mothers is not explained, nor is the discrepancy between maternal and neonatal genotoxicity explored in depth.
Response 3: Thank you for the observation, we added the next paragraph in the discussion section on lines 244- 246 “This behavior may be attributed to the fact that newborns still possess immature xenobiotic elimination systems, and hepatic enzyme activity begins only days after birth, which delays the clearance of foreign compounds”.
Comments 4: The discussion lacks the depth and scientific rigor necessary to interpret your findings in the context of existing literature.
Response 4: As we mentioned in the discussion section, no previous studies have assessed the genomic-level biosafety of Hs in Wistar rats and their offspring following transplacental exposure. Therefore, our findings cannot be directly compared with existing literature. Nonetheless, based on our results, consumption of ≥ 4.85 g/day of Hs during pregnancy may induce genotoxic and cytotoxic effects in the offspring due to transplacental exposure. Similarly, maternal intake of 19.4 g/day throughout gestation may lead to cytotoxic effects in the mother, we added this results in lines 336-343
Comments 5: The manuscript has only undergone superficial edits, while the core scientific and methodological limitations persist.
Response 5: Thank you for your observation. We have now implemented the suggested changes regarding the statistical analysis, methodology, and the presentation of results. We would greatly appreciate any further guidance or specific suggestions you may have on how we could continue to improve the quality of our manuscript.
Comments 6: The relationship between genotoxicity and teratogenicity is completely omitted.
Response 6: Thank you for pointing that, we added the next information in the main text in the discussion section on lines 224-230 “Genotoxicity and teratogenicity are mechanistically independent yet closely related processes, as genetic damage, particularly in germ cells or during early organogenesis, can compromise genomic integrity and trigger apoptosis or defective repair. Such alterations impede the proliferation, migration, and differentiation of cells essential for morphogenesis, leading to teratogenic outcomes ranging from minor anomalies to severe malformations or embryonic loss. Therefore, preserving the stability and proper functioning of the genome is critical to ensuring normal embryonic and fetal development. National Research Council (2000). Mechanisms of Developmental Toxicity.”
Comments 7: The methods of euthanasia are unethical.
Response 7: We used ether as anesthetic because in Mexico, the animal sacrifice is governed by Mexican Official Standard NOM-033-SAG/ZOO-2014, Methods for killing domestic and wild animals. On section 11, entitled Prohibited Methods, establishes the following: “Any method or substance not authorized in this list is prohibited: The use of substances that induce muscle paralysis without causing loss of consciousness and that cause death by asphyxiation is prohibited. These substances are: succinylcholine, curare, strychnine, galamine, and pancuronium, as well as their derivatives. The use of succinylcholine and curare is only authorized in crocodiles and megavertebrates in emergency cases; The use of potassium chloride (KCl) in any form to cause the death of animals is also prohibited, as its administration causes intense pain and anxiety, followed by cardiac arrest in diastole in the conscious individual. Its use is only authorized for megavertebrates, provided that the animal is under deep anesthesia and this is verified by a veterinarian; The use of hypothermia and electricity is prohibited for stunning, anesthesia, slaughter, and euthanasia of all reptiles; It is prohibited to kill rodents, lagomorphs, and small mammals by hypothermia and/or freezing, by chest compression, by strangulation, by drowning, by air embolism, or by other mechanical methods of asphyxiation”. Since ether is not among the methods prohibited by Mexican regulations, and in accordance with the regulations governing the operation of the rodent vivarium at the Autonomous University of Zacatecas which says that the recommended inhalable anesthetics for euthanasia of animals weighing less than 7 kg, or in those in which venipuncture is difficult, are: halothane, enflurane, isoflurane, methoxyflurane, and ether, alone or in combination with nitrous oxide.

Reviewer 2 Report
Comments and Suggestions for Authors
All inquiries were addressed by authors
Author Response
Response to reviewer 2 comments (2nd round)
We sincerely thank you for the time and effort you have dedicated to reviewing our manuscript. We greatly appreciate your insightful comments and constructive feedback. We will carefully follow your recommendations to improve the clarity, structure, and overall quality of the manuscript. Your observations have been invaluable in helping us enhance the scientific rigor and presentation of our work, and we remain committed to addressing each point thoroughly in our revised submission.

Reviewer 3 Report
Comments and Suggestions for Authors
Response 1
The authors cite the OECD guidelines, which is appropriate. However, they do not address the second part of the concern: contextualizing doses in relation to traditional human use (e.g., teas, supplements). Add a sentence estimating how the doses compare to typical human intake (e.g., via body surface area scaling or average daily consumption), or clearly state that such a comparison is difficult due to differences in preparation and metabolism.
Response 2
The response merely repeats the reviewer’s criticism without addressing or rebutting it. No attempt to justify the chosen window or acknowledge the limitation.
Response 4
Reasonable explanation referencing established models. But authors should explicitly address litter effects (pseudo-replication), which the response currently downplays.
Response 6
The authors address the use of normality tests and correct application of Kruskal–Wallis, but still mention using repeated-measures ANOVA, which is not appropriate for non-parametric or small-sample data, unless clearly justified. Clarify that repeated-measures ANOVA was only applied to data that met parametric assumptions. If not, replace with Friedman test or paired Wilcoxon as needed.
Response 7
The authors provide spectrophotometer settings but do not address the specificity limitation of the TBA-MDA assay.
Response 8
Referring to food labeling standards does not substitute for botanical authentication or a voucher specimen.
Response 10
While the authors acknowledge limitations, the response is vague and does not address dose extrapolation or risk-benefit relevance.
DISCUSSION
Genotoxicity findings are repeated with minor variations. Consolidate the interpretation of genotoxicity in a single coherent paragraph, clearly distinguishing maternal and neonatal effects.
The use of phrases like “this response does not follow a typical dose-dependent pattern” should be more analytically framed. Explicitly name the hormetic model and relate it to known toxicology frameworks, e.g., “U-shaped” or “biphasic response.”
While MDA generation is well described, the mechanistic link between oxidative stress and genotoxicity or cytotoxicity is not fully developed. Add a sentence connecting oxidative stress to DNA damage pathways (e.g., ROS-induced strand breaks, lipid peroxidation intermediates).
Fix missing commas, awkward constructions, and unnecessary phrases.
MATERIALS AND METHODS
Redundant or clunky phrasing:
- “Subsequently dried… and subsequently mechanically pulverized…” → simplify to “then dried… and pulverized…”
Numerous run-on or awkwardly long sentences (e.g., lines 338–344, 373–379, 389–392).
Inconsistent tense and phrasing throughout. Example: “we obtained the dry powder…” vs. passive voice used elsewhere.
Mesh No. 60 corresponds to ~250 μm, not 255–355 nm. This is likely a unit typo.
It is not clear whether the 79.3:20:0.7 solvent mix is volume/volume (v/v). Specify.
Buffer composition for tissue homogenization (e.g., pH, presence of protease inhibitors?) should be specified.
Clarify MDA calibration range and whether it was linear; mention if values were normalized to protein concentration.
Phytochemical analysis is referenced as "[64]" but not summarized at all in this section. A short mention of major compound classes (e.g., anthocyanins, flavonoids) would be helpful for context.
Author Response
Response to reviewer 3 comments (2nd round)
We sincerely thank you for the time and effort you have dedicated to reviewing our manuscript. We greatly appreciate your insightful comments and constructive feedback. We will carefully follow your recommendations to improve the clarity, structure, and overall quality of the manuscript. Your observations have been invaluable in helping us enhance the scientific rigor and presentation of our work, and we remain committed to addressing each point thoroughly in our revised submission.
Comments 1: The authors cite the OECD guidelines, which is appropriate. However, they do not address the second part of the concern: contextualizing doses in relation to traditional human use (e.g., teas, supplements). Add a sentence estimating how the doses compare to typical human intake (e.g., via body surface area scaling or average daily consumption), or clearly state that such a comparison is difficult due to differences in preparation and metabolism.
Response 1: Thank you for the advice, we added the next paragraph in the main text on lines 336-343 “Using the standard body surface area conversion proposed by the Food and Drug Administration (FDA), the tested doses were 500, 100, and 2000 mg/kg, equivalent to approximately 81, 162, and 324 mg/kg in humans, or approximately 4.85, 9.7, and 19.4 g/day for a 60 kg person. Typical consumption of Hs infusions varies from 1 to 10 g/day, depending on the preparation. Therefore, according to the results of this study, during pregnancy, doses of ≥ 4.85 g/day should not be consumed due to their genotoxic and cytotoxic impact on the newborn after transplacental exposure, similarly, maternal intake of 19.4 g/day throughout gestation may lead to cytotoxic effects in the mother [65].”.
Comments 2: The response 2 merely repeats the reviewer’s criticism without addressing or rebutting it. No attempt to justify the chosen window or acknowledge the limitation.
Response 2: I am sorry for that; it was a mistake. The different compounds were administered only during the last 5 days of gestation because, after exposure to EHHs, any chromosomal damage in erythrocytes becomes visible in the PCEs that appear in peripheral blood approximately 24–48 h after the injury and stabilizes in circulating erythrocytes at 72 h. Sampling during the last 5 days ensures that sufficient time has elapsed between the first day of administration and the final sample for micronuclei to form and be detected, this information was added to main text in lines 425-430.
Comments 3: (Response 4) Reasonable explanation referencing established models. But authors should explicitly address litter effects (pseudo-replication), which the response currently downplays.
Response 3: The selection of six newborns per litter was carried out through simple randomization (three males and three females), following the methodology established by Gómez-Meda et al. (2004) “Folate supplementation of cyclophosphamide-treated mothers diminishes micronucleated erythrocytes in peripheral blood of newborn rats. Environmental and Molecular Mutagenesis, 44(2), 174–178. https://doi.org/10.1002/em.20037.” In our study, as well as in reports from other authors, intra-litter variability has been observed in the evaluated parameters. However, we acknowledge that selecting six neonates per litter may artificially increase the sample size, potentially leading to biased or misleading conclusions.
Comments 4: The authors address the use of normality tests and correct application of Kruskal–Wallis, but still mention using repeated-measures ANOVA, which is not appropriate for non-parametric or small-sample data, unless clearly justified. Clarify that repeated-measures ANOVA was only applied to data that met parametric assumptions. If not, replace with Friedman test or paired Wilcoxon as needed.
Response 4: To assess the genotoxicity and cytotoxicity of a xenobiotic using the micronucleus assay, it is necessary to compare the means of the dependent variable (MN, EPCMN, or EMN) at different time points (24, 48, 75, 96, and 120 hours) relative to the baseline value. For this reason, a repeated-measures ANOVA is employed, as each animal essentially serves as its own control.
Comments 5: The authors provide spectrophotometer settings but do not address the specificity limitation of the TBA-MDA assay.
Response 5: We added the next paragraph talking about the possible limitations of the MDA quantification assay in lines 300-304. “Although the assay used for MDA quantification has certain limitations, such as cross-reactivity with other aldehydes and carbonyl compounds MDA remains a widely accepted biomarker for assessing lipid peroxidation. Moreover, it is a commonly employed method, which facilitates comparison of our results with previous literature. Finally, to minimize bias, the appropriate control groups were included”
Comments 6: Referring to food labeling standards does not substitute for botanical authentication or a voucher specimen.
Response 6: Thank you for your comment. We agree that labeling regulations do not replace proper botanical authentication of the plant. For this reason, we contacted Sam’s Club customer service to request more detailed information regarding the plant species. They informed us that the Mexican varieties of Hibiscus sabdariffa include Alma Blanca, Rosalíz, Cotzaltzin, and Tecoanapa, and that the variety they work with is Rosalíz. Therefore, the plant material used in our study corresponds to the Rosalíz variety, we added this information on line 368
Comments 7: (Response 10) While the authors acknowledge limitations, the response is vague and does not address dose extrapolation or risk-benefit relevance.
Response 7: Thank you for pointing that, we added the next paragraph where we extrapolate our results to human contexts in lines line 336-343 “Using the standard body surface area conversion proposed by the Food and Drug Administration (FDA), the tested doses were 500, 100, and 2000 mg/kg, equivalent to approximately 81, 162, and 324 mg/kg in humans, or approximately 4.85, 9.7, and 19.4 g/day for a 60 kg person. Typical consumption of Hs infusions varies from 1 to 10 g/day, depending on the preparation. Therefore, according to the results of this study, during pregnancy, doses of ≥ 4.85 g/day should not be consumed due to their genotoxic and cytotoxic impact on the newborn after transplacental exposure.
Comments 8: (Discussion section) genotoxicity findings are repeated with minor variations. Consolidate the interpretation of genotoxicity in a single coherent paragraph, clearly distinguishing maternal and neonatal effects.
Response 8: We modified in lines 224-247 the redaction of that section to make it clearer, describing first the results from controls and then the results obtained from the EHHs “Our results demonstrated that exposure to CP induced myelosuppression, as evidenced by the decrease in the proportion of PCEs. In addition, short-term genotoxic effects were observed, reflected by an increase in the proportion of MNPCEs in both the mothers and their newborns, as well as long-term genotoxic effects, indicated by an increase in the proportion of MNEs in the newborns [34,40,41,42]. The administration of EHHs during gestation in Wistar rats did not induce genotoxicity, however, the 2,000 mg/kg dose did cause cytotoxicity at 120 hours, as evidenced by a reduction in the number of PCEs. This decrease is associated with a myelosuppressive effect on the bone marrow. Newborns from mothers exposed to EHHs exhibited short-term genotoxicity, reflected by an increase in the proportion of MNPCEs at all three evaluated doses (500, 1,000, and 2,000 mg/Kg), and long-term genotoxicity at the 500 mg/Kg dose. This response does not follow a typical dose-dependent pattern, suggesting the possible presence of a non-monotonic dose–response relationship. This behavior may be attributed to the fact that newborns still possess immature xenobiotic elimination systems, and hepatic en-zyme activity begins only days after birth, which delays the clearance of foreign com-pounds [57,58].The fact that EHHs exhibited a myelostimulant effect despite existing evidence of their cytotoxicity supports the current understanding of the nature of hematopoietic stem cells, which are believed to possess a unique mechanism of self-renewal and differentiation [59].
Comments 9: The use of phrases like “this response does not follow a typical dose-dependent pattern” should be more analytically framed. Explicitly name the hormetic model and relate it to known toxicology frameworks, e.g., “U-shaped” or “biphasic response.”
Response 9: We added the next information in the paragraph where we talk about the hormetic effect in lines 323-324 “This behavior could be associated with a “U” shaped response curve.”
Comments 10: While MDA generation is well described, the mechanistic link between oxidative stress and genotoxicity or cytotoxicity is not fully developed. Add a sentence connecting oxidative stress to DNA damage pathways (e.g., ROS-induced strand breaks, lipid peroxidation intermediates).
Response 10: We added the next information about the DNA damage caused by MDA generation in lines 291-298 “ROS can form covalent DNA adducts, induce single and double-strand breaks, modify nitrogenous bases (for example, through the formation of 8-oxoguanine), promote depurination or depyrimidination, and disrupt the integrity of phosphodiester bonds. All of these processes can compromise the fidelity of DNA replication and transcription, increase the likelihood of mutations, and activate pathways related to apoptosis or DNA repair. Collectively, these mechanisms establish oxidative stress as a key contributor to genotoxicity and to the onset of various pathologies, including cancer and degenerative or metabolic diseases”
Comments 11: Fix missing commas, awkward constructions, and unnecessary phrases.
Response 11: Thank you for pointing that, we checked the whole text to correct those mistakes
Comments 12: “Subsequently dried… and subsequently mechanically pulverized…” → simplify to “then dried… and pulverized…”
Response 12: We corrected the paragraph redaction to fix those mistakes “The calyxes were dried in an oven at less than 50° C and mechanically pulverized” (Lines 373-374)
Comments 13: Numerous run-on or awkwardly long sentences (e.g., lines 338–344, 373–379, 389–392).
Response 13: Thank you for pointing that, we corrected the mistakes and repeated phrases in lines you mentioned
Comments 14: Inconsistent tense and phrasing throughout. Example: “we obtained the dry powder…” vs. passive voice used elsewhere.
Response 14: We checked the section to correct the mistakes in the redaction and reduce the redaction in passive voice (lines 367-393)
Comments 15: Mesh No. 60 corresponds to ~250 μm, not 255–355 nm. This is likely a unit typo.
Response 15: We corrected that mistake an added the next information in line 375: “which was passed through a sieve to yield particles ~255 nm (mesh No. 60)”.
Comments 16: It is not clear whether the 79.3:20:0.7 solvent mix is volume/volume (v/v). Specify.
Response 16: We added “(v/v)” to specify the proportions used on line 378
Comments 17: Buffer composition for tissue homogenization (e.g., pH, presence of protease inhibitors?) should be specified.
Response 17: We specified the buffer’s pH in lines 460-461 “tissues were rinsed with phosphate buffer solution at pH 7.0”. Since we did no used any protease inhibitors, we did not declare that.
Comments 18: Clarify MDA calibration range and whether it was linear; mention if values were normalized to protein concentration.
Response 18: The calibration curve for MDA was constructed using concentrations ranging from 0 to 16 nmol, also, the calibration curve was linear within this range, with a correlation coefficient (R²) of 0.998. We added this information on lines 492- 494
Comments 19: Phytochemical analysis is referenced as "[64]" but not summarized at all in this section. A short mention of major compound classes (e.g., anthocyanins, flavonoids) would be helpful for context.
Response 19: We added the next paragraph in the main text on lines 391-393 “The phytochemical analysis revealed the presence of flavonoids, phenolic acids, and organic acids. Details of the phytochemical analysis conducted are available in the supplementary material”.

Round 3
Reviewer 3 Report
Comments and Suggestions for Authors
Comments 2 / Response 2
The authors added important timing information, but they still haven’t fully acknowledged or discussed the limitation of administering compounds only in the final 5 days of gestation. Add a line explicitly recognizing the limitation:
“We acknowledge that limiting exposure to the last 5 days of gestation may not capture earlier developmental effects, and this remains a limitation of our experimental design.”
Comments 3 / Response 3
While the authors cite literature and acknowledge intra-litter variability, the concern about pseudo-replication is not directly addressed.
Explicitly state how litter effects were handled statistically or acknowledge that analysis was not adjusted for litter as the experimental unit.
Comments 4 / Response 4
The authors justify repeated-measures ANOVA without addressing the concern: Was it used only if data met parametric assumptions? If not, it’s statistically inappropriate.
Comments 6 / Response 6
Contacting the supplier is useful, but this doesn't substitute for botanical authentication or voucher deposition.
Comments 7 / Response 7
The response overlaps with the earlier dose extrapolation and does not add new substance regarding risk–benefit relevance.
Comment 15 / Response 15
The authors replaced one error with another. Mesh No. 60 corresponds to ~250 μm, not 255 nm.
Comment 17 / Response 17
The authors specified pH but not whether protease inhibitors were deliberately excluded.
Author Response
Response to reviewer 3 comments (3rd round)
We sincerely thank you for the time and effort you have dedicated to reviewing our manuscript. We greatly appreciate your insightful comments and constructive feedback. We will carefully follow your recommendations to improve the clarity, structure, and overall quality of the manuscript. Your observations have been invaluable in helping us enhance the scientific rigor and presentation of our work, and we remain committed to addressing each point thoroughly in our revised submission.
Comments 1: The authors added important timing information, but they still haven’t fully acknowledged or discussed the limitation of administering compounds only in the final 5 days of gestation. Add a line explicitly recognizing the limitation: “We acknowledge that limiting exposure to the last 5 days of gestation may not capture earlier developmental effects, and this remains a limitation of our experimental design.”
Response 1: Thank you for that suggestion, we added the next paragraph “We acknowledge that limiting exposure to the last 5 days of gestation may not capture earlier developmental effects, and this remains a limitation of our experimental design” to main text on lines 444-446
Comments 2: (Response 3) While the authors cite literature and acknowledge intra-litter variability, the concern about pseudo-replication is not directly addressed. Explicitly state how litter effects were handled statistically or acknowledge that analysis was not adjusted for litter as the experimental unit.
Response 2: Thank you for that suggestion, we acknowledge that the statistical analysis was not adjusted for litter as the experimental unit. Individual pups were used as data points, and litter effects were not explicitly accounted for in the analysis. We recognize this as a limitation of the study and we added a statement to this effect in the Discussion section in lines 367-369 “Future studies should consider using the litter as the unit of analysis or apply statistical models that incorporate litter effects to avoid potential pseudo-replication”.
Comments 3: (response 4) The authors justify repeated-measures ANOVA without addressing the concern: Was it used only if data met parametric assumptions? If not, it’s statistically inappropriate.
Response 3: We used the repeated measures ANOVA test for non-parametric data because, according to Test No. 474 “OECD GUIDELINE FOR THE TESTING OF CHEMICALS Mammalian Erythrocyte Micronucleus Test” does not impose a single test, but refers to the “literature” section (references 44–47 of the document itself) to select methods appropriate to the design and distribution of the data. Among the most cited are tests and analysis of variance (ANOVA). Furthermore, according to the test design, each animal must be compared with its respective baseline value (0 h), and the repeated measures ANOVA test allows us to make this type of comparison. Also, similar studies use the ANOVA test to make the statistical analysis, for example: https://doi.org/10.1155/2022/955401 and https://doi.org/10.1016/j.yrtph.2017.03.017
Comments 4: (Response 6) Contacting the supplier is useful, but this doesn't substitute for botanical authentication or voucher deposition.
Response 4: We acknowledge that formal botanical authentication by a taxonomist was not performed, nor was a specimen deposited in an herbarium. Since this procedure was not included in the original protocol so we added the following clarification to asses that limitation :“This study used plant material certified by the supplier, but did not include botanical authentication by a specialist. For future studies, we recommend conducting a thorough morphological comparison.” On lines 385-387
Comments 5: (Response 7) The response overlaps with the earlier dose extrapolation and does not add new substance regarding risk–benefit relevance.
Response 5: Thank you for that observation, we added the next paragraph in the discussion section on lines 345-356 “Although well‐defined safety thresholds for medicinal plant use during pregnancy are lacking, it is essential to conduct studies assessing both acute and chronic exposure and their impact on fetal development to establish a reliable safety profile. In our preclinical model, administration of the EHHs led to elevated markers of cytotoxicity in the mothers and both cytotoxic and genotoxic effects in the offspring following transplacental exposure, underscoring a clear risk profile. Consequently, despite the well‐documented antihypertensive, antioxidant, and lipid‐improving benefits of Hs anthocyanins and phenolics in humans, it is crucial to define the toxicological margin. Given the potential for diuretic and metabolic interactions and the absence of data on placental biodistribution, we recommend advancing to Phase I/II clinical trials in healthy women and low‐risk pregnant subjects to characterize tolerability, pharmacokinetics, placental bioavailability, and to establish intake limits that maximize benefit without compromising maternal–fetal safety”
Comments 6: (Response 15) The authors replaced one error with another. Mesh No. 60 corresponds to ~250 μm, not 255 nm.
Response 6: Thank you for pointing that, we corrected that mistake on lines 389
Comments 7: (Response 17) The authors specified pH but not whether protease inhibitors were deliberately excluded.
Response 7: We decided not to use protease inhibitors because the phosphate buffer was used exclusively to rinse the tissues prior to homogenization, with the aim of removing blood residues and minimizing interference in the quantification of malondialdehyde. Since the washing did not involve cell lysis or prolonged exposure to the environment, it was not necessary to add protease inhibitors. We added this justification to main text on lines 477-481

Round 4
Reviewer 3 Report
Comments and Suggestions for Authors
The authors have thoroughly addressed all comments and concerns raised during this and all previous rounds of review. They have made appropriate revisions, acknowledged the study's limitations, and provided sufficient clarification and justification where needed. The manuscript now meets the scientific and ethical standards expected for publication. I recommend acceptance pending minor English language and editorial corrections.